# Motor domain phosphorylation increases nucleotide exchange and turns MYO6 into a faster and stronger motor

Janeska J. de Jonge[1,8], Andreas Graw[2,3,8], Vasileios Kargas[1,4,5], Christopher Batters [1,2,3], Antonino F. Montanarella [2,3], Tom O'Loughlin[1], Chloe Johnson[1], Susan D. Arden[1], Alan J. Warren [1,4,5], Michael A. Geeves [6], John Kendrick-Jones[7], Nathan R. Zaccai[1], Markus Kröss [2,3], Claudia Veigel [2,3] ✉ & Folma Buss [1] ✉

Myosin motors perform many fundamental functions in eukaryotic cells by providing force generation, transport or tethering capacity. Motor activity control within the cell involves on/off switches, however, few examples are known of how myosins regulate speed or processivity and fine-tune their activity to a specific cellular task. Here, we describe a phosphorylation event for myosins of class VI (MYO6) in the motor domain, which accelerates its ATPase activity leading to a 4-fold increase in motor speed determined by actin-gliding assays, single molecule mechanics and stopped flow kinetics. We demonstrate that the serine/threonine kinase DYRK2 phosphorylates MYO6 at S267 in vitro. Single-molecule optical-tweezers studies at low load reveal that S267-phosphorylation results in faster nucleotide-exchange kinetics without change in the working stroke of the motor. The selective increase in stiffness of the acto-MYO6 complex when proceeding load-dependently into the nucleotide-free rigor state demonstrates that S267-phosphorylation turns MYO6 into a stronger motor. Finally, molecular dynamic simulations of the nucleotide-free motor reveal an alternative interaction network within insert-1 upon phosphorylation, suggesting a molecular mechanism, which regulates insert-1 positioning, turning the S267-phosphorylated MYO6 into a faster motor.

Myosin motor proteins are actin-activated ATPases that produce force and movement following a highly conserved mechanochemical cycle[1,2]. Their biological roles are highly diverse, requiring fine-tuning and adaptations to their kinetic cycle, which is a succession of ATP-hydrolysis, tightly coordinated with actin-binding and lever arm movement[3]. All parts of the kinetic cycle are tuneable, but myosins show in particular variation in the rate of ADP release, a critical step that determines not only the velocity but also the duty ratio of the

[1]Cambridge Institute for Medical Research, Department of Clinical Biochemistry, University of Cambridge, Cambridge Biomedical Campus, The Keith Peters Building, Hills Road, Cambridge CB2 0XY, UK. [2]Department of Cellular Physiology, Biomedical Centre (BMC), Ludwig-Maximilians-Universität München, Grosshadernerstrasse 9, 82152 Planegg-Martinsried, Germany. [3]Centre for NanoScience (CeNS), Ludwig-Maximilians-Universität München, Schellingstrasse 4, 80799 München, Germany. [4]Wellcome MRC Cambridge Stem Cell Institute, University of Cambridge, Cambridge, UK. [5]Department of Haematology, University of Cambridge, Cambridge, UK. [6]School of Biosciences, University of Kent, Canterbury, UK. [7]MRC Laboratory of Molecular Biology, Francis Crick Avenue, Cambridge Biomedical Campus, Cambridge CB2 0QH, UK. [8]These authors contributed equally: Janeska J. de Jonge, Andreas Graw. ✉e-mail: Claudia.Veigel@med.lmu.de; fb207@cam.ac.uk

motor, which is the fraction of time the motor is attached to an actin filament during one complete chemo-mechanical cycle[4].

Myosins of class VI (MYO6) contain a unique, class-specific 53-amino-acid-insert (insert-2) at the junction between the converter domain and the canonical calmodulin-binding IQ motif that repositions the lever arm and causes MYO6 to move towards the minus end of actin filaments, in the opposite direction to other myosin family members[5,6]. MYO6 also contains an additional 26-amino-acid-insert (insert-1) in the motor domain, close to the nucleotide binding pocket[5]. Deletion of insert-1 increases the rate of ATP-binding, possibly by removing a steric restriction on the access of ATP to its site by controlling the position of leucine 310 at the entrance of the nucleotide binding pocket[7].

The unique directionality of MYO6 may explain the large variety of cellular roles associated with this myosin ranging from plasma membrane anchoring in stereocilia, to regulation of vesicle transport and actin filament dynamics[8–10]. This functional diversity is facilitated by a host of binding partners associated with the C-terminal tail[11]. These cargo-adaptor proteins not only regulate the spatial and temporal targeting of MYO6 within the cell, but also appear crucial for motor activation and transition between monomeric and oligomeric states[12–17]. Additional regulation may be provided by $Ca^{2+}$ that relieves a folded autoinhibited conformation of MYO6 and promote cargo binding in vitro[18]. Further fine-tuning via phosphorylation in the motor and tail domains might constitute reversible modification in response to local environmental cues[19–21]. MYO6 is indeed phosphorylated at T405 in the motor domain in proximity to the actin-binding interface[22]. This modification, when tested in vitro, appears to have limited impact on the overall motor activity[23]. Cellular studies, however, suggest that T405 phosphorylation influences endosome distribution and recruitment of MYO6 to sites of Salmonella invasion[22,24]. Despite this progress, the means by which the activity and diverse functions of MYO6 are coordinated and regulated in cells, remain unclear.

Here, we describe the effect of a single phosphorylation event in the catalytic domain that controls the actin-activated MYO6-ATPase activity and force generation of this motor. Using phospho-proteomics we identified a serine phosphorylation site at amino acid 267 in close proximity to insert-1. Analysing the effect of this phosphorylation event on the mechanics and kinetics of MYO6 in vitro shows accelerated ATPase activity for phosphomimetic MYO6[S267E]. Increased ADP-release and ATP-binding rates result in a faster, lower duty ratio motor that translocates F-actin in vitro at low load with a 4-fold increase in velocity, without changing the size of the working stroke. The kinase responsible for phosphorylating MYO6 at S267 is DYRK2, a crucial regulator of many different cellular processes. The importance of S267-phosphorylation is also being investigated in a cell-based assay. Using a mutant, plus-end directed MYO6+ that induces formation of filopodia-like protrusions at the cell surface[25], we demonstrate that expression of MYO6+[S267E] increases the number of filopodia at the cell surface by 100%. Molecular dynamic (MD) simulations of the detached, nucleotide-free motor domain show S267-phosphorylation to change the local H-bonding network around insert-1. These changes in the local dynamics around insert-1 also alter the conformation of the adjacent L310 loop in front of the ATP-binding pocket, which in turn may lead to greater accessibility allowing faster ADP-release and ATP-binding. Single-molecule and stopped-flow experiments with phosphomimetic MYO6[S267E] reveal faster nucleotide exchange with load-dependent transition kinetics for both ADP-release and ATP-binding. Furthermore, we observe increased stiffness of the strong-binding acto-MYO6[S267E] complex following ADP-release and preceding ATP binding, showing that the nucleotide-free acto-MYO6[S267E] complex can generate 40% more force than acto-MYO6[WT] and demonstrating that phosphorylation at S267 turns MYO6 into a stronger motor protein.

These results illustrate that a single phosphorylation event in the motor domain can modulate actin-activated ATPase activity and

change the mechanical properties of the force generating acto-MYO6 complex, converting MYO6 at low load from a slower, higher duty ratio motor to a faster, lower duty ratio motor that generates more force.

## Results

### Identification of a MYO6 phosphorylation site at S267 in the motor domain

To identify regulatory phosphorylation in MYO6, we immuno-precipitated endogenous MYO6 from epidermal cancer A431 cells or pulled-down GFP-tagged, full-length MYO6 from retinal pigment epithelial (hTERT-RPE1) cells and analysed the precipitates for phospho-peptides by mass spectrometry (Fig. 1a). Sequence coverage was 70–90% for each sample ($n = 4$) and identified a phosphopeptide covering the area containing T405, a known MYO6 phosphorylation site[22,23], and a phosphopeptide containing a serine at positions 266 and 267 in the motor domain, with serine 267 displaying far greater site localisation probability relative to its neighbour (Fig. 1b, c). The position of S267 close to insert-1 (A) and the position of T405 at the actin binding interface (B) is shown in the predicted MYO6 *AlphaFold* structure (Fig. 1d). The S267 phosphorylation site is conserved across vertebrates but is not present in insects or worms (Fig. 1e). We next analysed the phosphorylation status of T405 and S267 in response to external stimuli such as epidermal growth factor (EGF) or phorbol 12-myristate 13-acetate (PMA). Our results demonstrate that T405 is phosphorylated in 75% of the peptides, while S267 shows low levels of phosphorylation (1–2%) when the cells are starved, highlighting the potentially transient nature of S267 phosphorylation (Fig. 1f, g). S267 phosphorylation may also occur only in a limited subset of MYO6 within the cell associated with a specific task. Notably, both sites showed moderate increases in phosphorylation upon stimulation with EGF or PMA (Fig. 1f, g).

### DYRK2 phosphorylates MYO6 at S267

To identify the kinase responsible for S267 phosphorylation, we used a combination of prediction tools based on various algorithms to select 31 kinases with the highest probability of phosphorylating the SSP sequence. A radioactive HotSpot™ kinase assay using this kinase panel together with the MYO6 peptide[26,27] revealed that dual-specificity tyrosine phosphorylation-regulated kinase 2 (DYRK2) was the only kinase able to phosphorylate the MYO6 peptide[26,27] (Fig. 2a). DYRK2 belongs to the CMGC kinase superfamily that can phosphorylate both tyrosine and serine/threonine residues. It is involved in a wide range of biological functions, including cell survival, development, proteasomal regulation and microtubule formation[26,27]. To further verify that DYRK2 is the relevant kinase for the S267 site, we tested recombinant DYRK2 (Fig. 2b) on different MYO6 peptides (Fig. 2d) using the ADP-glow assay (Fig. 2c). We showed that DYRK2 specifically phosphory-lated a peptide containing the SSP or the ASP consensus sequence, but not one in which S267 was replaced by alanine (SAP) or in which both S266 and S267 were mutated (AAP) (Fig. 2d). Importantly, DYRK2 also phosphorylated full-length MYO6, either purified from insect cells (Fig. 2e) or immunoprecipitated from A431 cells (Fig. 2f). No phos-phorylation of T405-containing peptides by DYRK2 was observed (Fig. 2g). Taken together these results demonstrate that DYRK2 is the kinase specifically phosphorylating MYO6 at S267 in vitro.

### Phosphomimetic MYO6[S267E] moves with higher velocity and reduced processivity on actin filaments

S267 is in close proximity to the unique insert-1 of MYO6, which has previously been suggested to modulate nucleotide affinity and ATPase activity[7]. Therefore, we first determined the effect of S267 phosphory-lation on MYO6 motor activity and velocity, using in vitro actin gliding assays to measure the translocation speed of actin filaments by surface-immobilised full-length no insert wildtype (WT) and mutant MYO6 (Fig. 3a). While non-phosphorylatable MYO6[S267A] and MYO6[WT]

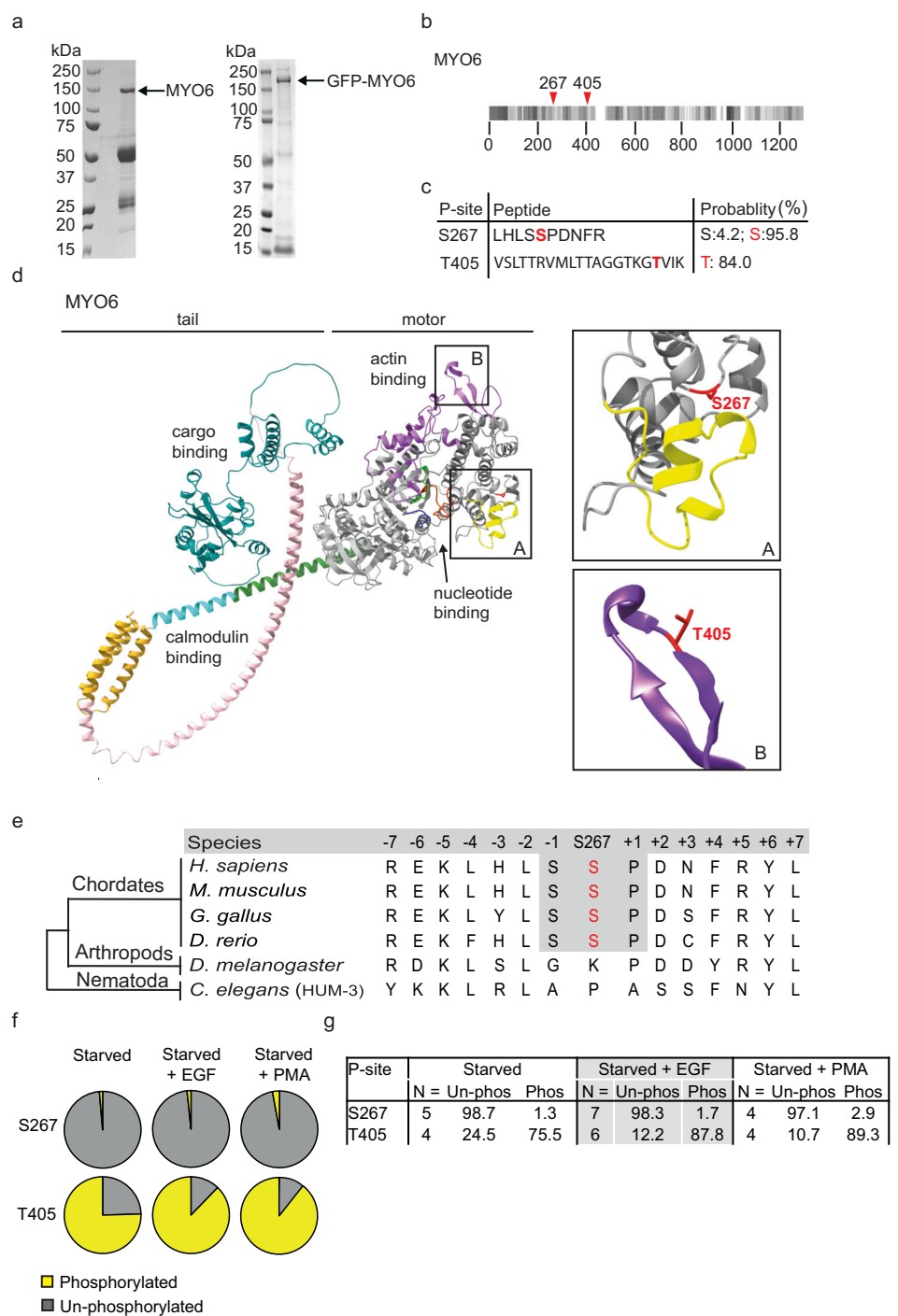

**Fig. 1 | DYRK2 phosphorylates MYO6 at S267 in the motor domain in vitro.**
**a** SDS-PAGE of GFP-tagged MYO6 immunoprecipitated from RPE cell lysate using
GFP nanobodies (right) and endogenous MYO6 immunoprecipitated from A431 cell
lysate using MYO6 antibodies (left). A representative image is shown of the
experiment performed more than 10 times. The arrows indicate the position of
GFP-MYO6 or endogenous MYO6. **b** MYO6 sequence peptide coverage by mass
spectrometry. Identified peptides shown in shades of grey. **c** Sequence of the
identified phosphopeptide LHLSSPDNFR for the S267 and
VSLTTRVMLTTAGGTKGTVIK for the T405 and site probabilities for S266 and S267
or T405. **d** Predicted *AlphaFold* (v2.0) structure of MYO6: motor domain (grey),
neck domain (green/turquoise), neck domain extension (gold), SAH domain (pink)

and C-terminal cargo-binding domain (blue). The inset A shows the position of S267
(red) and insert-1 (yellow) and inset B highlights the position of T405 (red) in the
surface loop (purple) at the actin binding site. **e** Conservation of MYO6 at S267
across species. **f** Pie charts illustrating the percentages of S267 or T405 phos-
phorylation in MYO6 immunoprecipitated from A431 cells grown in serum-free
media (starved) or after stimulation with EGF for 15 min or phorbol 12-myristate 13-
acetate (PMA) for 20 min. **g** Table summarising the percentages of S267 or T405
phosphorylation for MYO6 immunoprecipitated from A431 cells grown in serum-
free media (starved) or after stimulation with EGF for 15 min or phorbol 12-
myristate 13-acetate (PMA) for 20 min.

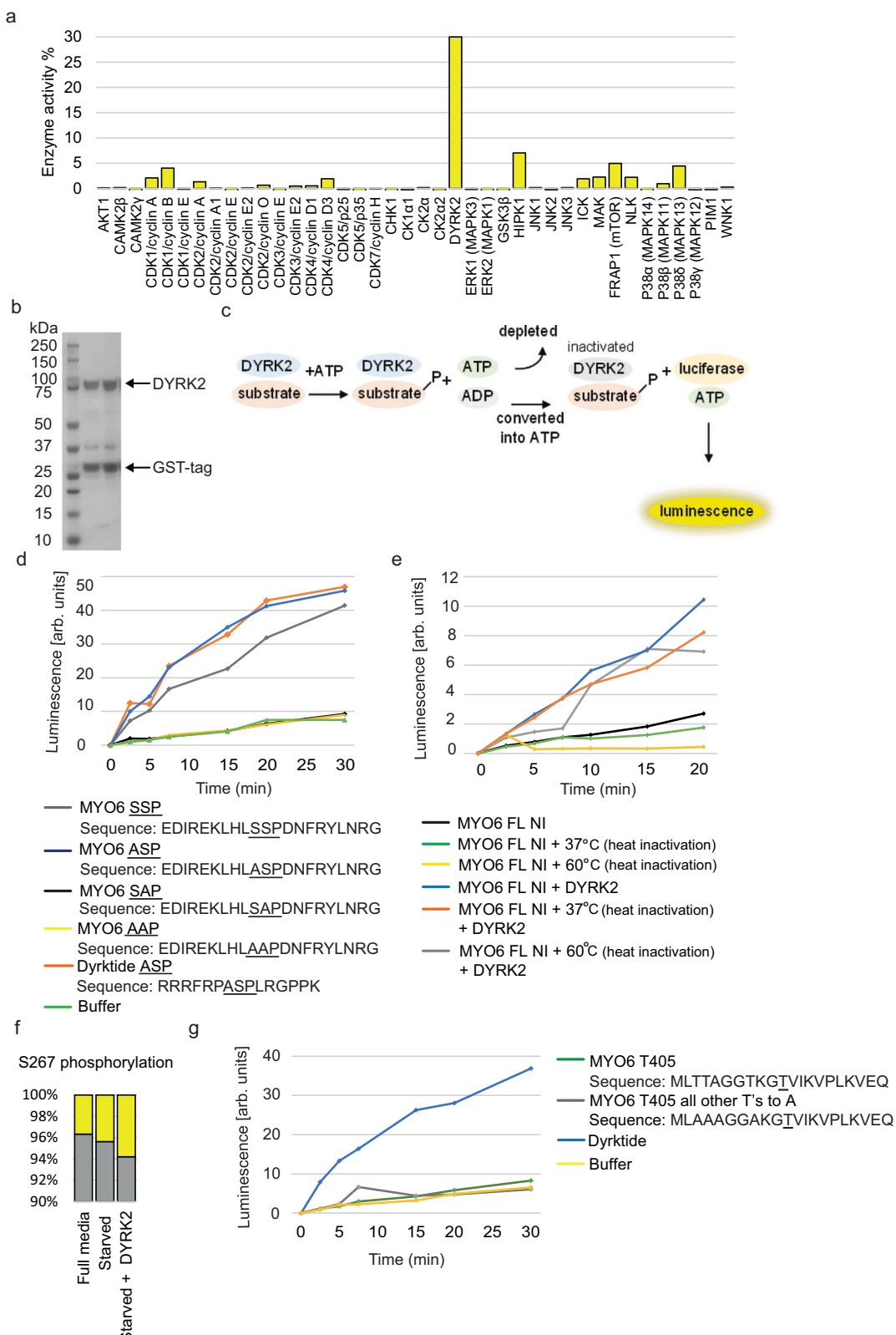

**Fig. 2 | DYRK2 specifically phosphorylates the MYO6 S267 peptide and the MYO6 full-length protein. a** Activities (HotSpot™ kinase assay) of 31 kinases with respect to the MYO6 peptide, compared with their ideal substrates. **b** A representative SDS-PAGE image of recombinant DYRK2 purified from *E. coli*. The purification has been performed several times. **c** Schematic overview of the ADP-glow assay used to measure DYRK2 activity. **d** ADP-glow™ kinase assays show that recombinant DYRK2 has ATPase activity with respect to both DYRK-specific substrate and wild-type MYO6 peptide containing the SSP and the ASP sequence, but not to peptides with the mutant SAP or AAP sequence. **e** DYRK2 shows ATPase activity towards the MYO6 motor domain in vitro. Before the ADP-glow assay full length MYO6 was heat-treated at 37 °C or 60 °C to inactivate endogenous ATPase activity that would interfere with the kinase assay. **f** MYO6 was immunoprecipitated from A431 cells grown in serum-free media (starved) and incubated with purified DYRK2 for 1 h before determining the level of S267 phosphorylation by mass spectrometry. **g** DYRK2 does not phosphorylate MYO6 at T405 using the wildtype MYO6 T405 peptide in the ADP-glow assay in vitro.

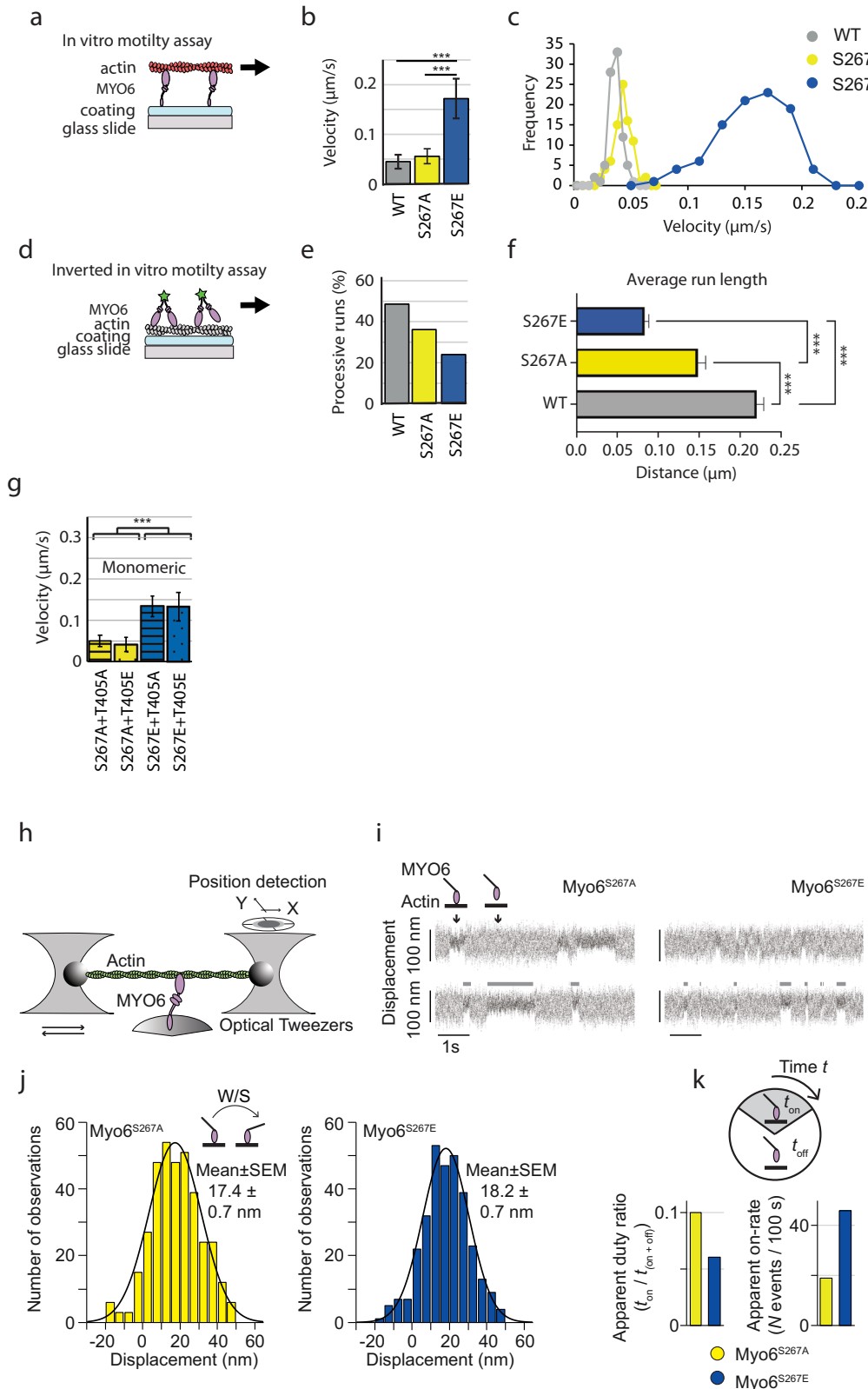

generated similar velocities (56 nm s⁻¹ and 45 nm s⁻¹ respectively), the phosphomimetic MYO6^S267E caused an almost 4-fold increase in speed (172 nm s⁻¹) (Fig. 3b, c). The ability of MYO6 to function as both a monomer and a dimer within cells indicates that a change in velocity could also alter the processivity of the dimeric motor. In an inverted in vitro motility assay, we quantified velocity, overall run-length and probability of processivity of leucine zipper-dimerised MYO6^WT,

MYO6^S267A or MYO6^S267E along fascin-stabilised, biotinylated actin filament bundles, immobilised via streptavidin on the surface of the experimental chamber (Fig. 3d). Whilst the average velocity of all three constructs was similar (≈240 nm s⁻¹), a significant difference in the probability of processive movement is observed, with only 24% of dimerised MYO6^S267E compared with 49% MYO6^WT or 36% MYO6^S267A molecules moving (Fig. 3e). Additionally, the median run-length of

**Fig. 3 | Increased actin-gliding velocity of MYO6$^{S267E}$ in vitro is achieved by faster nucleotide-exchange rates without change in the W/S of the motor.** **a** Scheme of the in vitro actin-gliding assay. Monomeric MYO6, immobilised on a nitrocellulose-coated glass surface, translocates rhodamine-phalloidin labelled F-actin in the presence of ATP. **b**, **c** Actin-gliding velocity (2 mM ATP, 22 °C) increased 4-fold from 45 nm s$^{-1}$ (MYO6$^{WT}$) and 56 nm s$^{-1}$ (MYO6$^{S267A}$) to 172 nm s$^{-1}$ for MYO6$^{S267E}$. 3 independent experiments from 3 protein purifications were performed; Statistical analysis was performed using an unpaired two-sided t-test. ***P < 0.01. MYO6$^{WT}$ n = 567, average 0.045 μm/s SD ± 0.008283, MYO6$^{S267A}$ n = 557, average 0.0557 μm/s SD ± 0.010488, MYO6$^{S267E}$ n = 569, average 0.1725 μm/s SD ± 0.0345.; **d** Scheme of the inverted in vitro motility assay. Dimerised MYO6 translocates on fascin-stabilised actin bundles. **e** Fraction of processive runs of MYO6$^{WT}$ (48.6%), MYO6$^{S267A}$ (36.2%) and MYO6$^{S267E}$ (24%). **f** Median run-length of MYO6$^{WT}$, MYO6$^{S267A}$ and MYO6$^{S267E}$ was determined in the inverted motility assay using 3 different flow cells for each protein. Statistical analysis was performed using the Kruskal-Wallis test by ranks, a non-parametric method for testing whether samples originate from the same distribution. It is used for comparing two or more independent samples of equal or different sample sizes. ***P < 0.01. MYO6$^{WT}$ n = 7926, average 0.22 μm SEM ± 0.0047, MYO6$^{S267A}$ n = 3981, average 0.148 μm SEM ± 0.0058 and MYO6$^{S267E}$ n = 2367, average 0.083 μm SEM ± 0.0061. **g** Actin-gliding velocity (2 mM ATP, 22 °C) for the double mutants MYO6$^{S267A/E}$ plus MYO6$^{T406A/E}$ were determined by the mutation at S267 and independent of the mutation at T405 (S267A+T406A 50 ± 14 nm s$^{-1}$; S267A+T405E 41 ± 17 nm s$^{-1}$; S267E+T405A 134 ± 25 nm s$^{-1}$; S267E+T405E 133 ± 35 nm s$^{-1}$ mean ± SD, N = 51 filaments for each double mutant, ***P < 0.01, unpaired two-sided t-test; 3 experiments from 3 different protein preparations, Fig. 4). **h** Scheme of single-molecule mechanical experiments using optical tweezers. **i** Raw data traces of single MYO6$^{S267A}$ and MYO6$^{S267E}$ molecules interacting with F-actin (single trap-stiffness $\kappa_{trap}$ ~ 0.02 pN nm$^{-1}$; 100 μM ATP). MYO6 binding events to actin were detected by changes in the variance of thermal motion. Grey bars indicate actin-attached dwell times. **j** To determine the working stroke (W/S) for the MYO6$^{S267A}$ and MYO6$^{S267E}$ the displacement distribution of actin-binding events was analysed at 100 μM ATP. **k** Characterisation of the apparent duty ratio and apparent on-rate of actin binding for the MYO6 A-and E-mutants. Source data are provided as a Source Data file.

MYO6$^{S267E}$ of 83 nm was significantly decreased compared with MYO6$^{S267A}$ (148 nm) and MYO6$^{WT}$ (220 nm) (Fig. 3f). The probability of binding followed by dissociation without processive movement indicated an alteration in the kinetics and potentially also in gating (coordination of the heads) of dimerised MYO6. Our results also reveal high levels of MYO6 phosphorylation at T405 (Fig. 1f) close to the actin-binding surface. To determine whether changes at the actin-binding interface affect MYO6 velocity, we generated a set of single and double mutants for MYO6 at T405 and S267, which were tested in the actin gliding assay. Our results indicate no additional effect of T405 phosphorylation on the S267-induced velocity change of MYO6 (Fig. 3g, Fig. 4).

### Single-molecule studies reveal faster nucleotide exchange for MYO6$^{S267E}$ without affecting the working stroke of the motor

The chemo-mechanical properties of MYO6$^{S267A}$- and MYO6$^{S267E}$-mutants were studied at the single molecule level using an optical tweezers apparatus in three-bead-configuration (Fig. 3h)[28]. We investigated whether S267 phosphorylation affected the amplitude of the working stroke (W/S) of the motor, its kinetics or both. Analysis of the binding events[29,30] (Fig. 3i) confirmed a similar W/S for both the MYO6$^{S267A}$- (17.4 ± 0.7 nm) and the MYO6$^{S267E}$-mutant (18.2 ± 0.7 nm) (Fig. 3j), consistent with previous data for MYO6$^{WT}$[31]. Actin-attached dwell-times $t_{on}$ for MYO6$^{S267A}$ were longer than for MYO6$^{S267E}$ (Fig. 3i, grey bars). While the apparent duty ratio (=$t_{on}/(t_{on} + t_{off})$) for MYO6$^{S267E}$ was reduced by a factor of about 2 compared with MYO6$^{S267A}$, the number of binding events per unit time increased only by the same factor, demonstrating that the real on-rate (=1/$t_{off}$) of actin-binding remained unchanged for both MYO6 mutants (Fig. 3k).

To assign the actin-attached dwell-times $t_{on}$ to specific biochemical states as the single motor proceeds through its ATPase cycle, we analysed $t_{on}$ with 10 and 100 μM ATP (Fig. 5a, b). The cumulative plots for the MYO6$^{S267A}$ and MYO6$^{S267E}$ mutants were described by two exponential components[32]. The ATP-independent component $k_1$ of 4.7 ± 0.6 s$^{-1}$ for MYO6$^{S267A}$ was increased by a factor of 4 to 19.4 ± 1.6 s$^{-1}$ for the MYO6$^{S267E}$, while the ATP-dependent rate $k_2$ only increased slightly (MYO6$^{S267A}$, 46 ± 5 mM$^{-1}$ s$^{-1}$; MYO6$^{S267E}$, 55 ± 5 mM$^{-1}$ s$^{-1}$). Both $k_1$ and $k_2$ for MYO6$^{S267A}$ agree closely with previous single-molecule data for MYO6$^{WT}$[31,33] and were confirmed for ATP concentrations differing by an order of magnitude. Ensemble-averaging analysis[34] enabled us to link $k_1$ and $k_2$ to specific mechanical events during a single motor's crossbridge cycle. In this experiment, $k_1$ is determined by the time required to complete both substeps of the W/S, while $k_2$ characterises the time between W/S 2 and detachment (Fig. 5a, c). Importantly, this analysis was consistent with the cumulative dwell-time analysis and links the chemo-mechanical data of the single-molecule experiments directly to structural studies of MYO6$^{WT}$[6] (Fig. 4) that also associate

ADP-release with W/S 2. Together, these single-molecule studies show that S267 phosphorylation increases both nucleotide-exchange rates $k_1$ and $k_2$ of actin-bound, force-generating MYO6 without changing the W/S of the motor, and quantitatively explain the S267 phosphorylation effect on in-vitro actin-gliding speed at low load, seen above.

### Solution kinetics highlights four-fold increase in ADP-release

In contrast to the single-molecule approach, solution kinetics studies using stopped-flow investigate specific biochemical transitions of the acto-myosin interaction in bulk, without the motors proceeding through a complete chemo-mechanical cycle. In these experiments, full-length no insert MYO6 binding to pyrene-labelled actin quenches the pyrene fluorescence so that, on ATP binding, the acto-MYO6 complex dissociates and the pyrene fluorescence increases (Fig. 5d)[35]. To determine ADP release, ADP was displaced from a pyrene-actin.MYO6.ADP complex by ATP excess. We observed a > 4-fold acceleration in ADP release for MYO6$^{S267E}$ (16 s$^{-1}$) compared with MYO6$^{S267A}$ (2.8 s$^{-1}$) and MYO6$^{WT}$ (3.5 s$^{-1}$) in the absence of calcium (Fig. 5e), consistent with the single-molecule data for $k_1$ described above. The difference in $k_1$ was smaller in the presence of calcium, with a 2-fold increase for MYO6$^{S267E}$ (5.9 s$^{-1}$) compared with MYO6$^{WT}$ (2.9 s$^{-1}$) (Supplementary Table 1). MYO6 has calmodulin light chains on the lever arm and so there is a question if calcium binding to the calmodulin plays any regulatory/modulatory role either as part of the on-off switch with cargo or via a change in the mechanical properties of the lever arm. Calcium binding to the first calmodulin of the lever arm of MYO1B and 1C has been reported to alter ADP release in an unloaded motor[36,37]. Since we have already shown that the phosphomimetic mutation can influence ADP release, it was of interest to examine if there is an interplay between calcium-calmodulin and ADP release. We find that MYO6$^{S267E}$ is inhibited by calcium (3–4 fold) and to a greater extent than MYO6$^{WT}$ or MYO6$^{S267A}$ (less than 50%). How this is linked to the previous reports that calcium plays a role in regulating cargo binding[18,38,39], remains to be explored.

The apparent second-order rate constant $k_2$ for ATP-induced dissociation of the acto-MYO6 complex was also determined for all three MYO6 mutants. The dissociation time courses followed single exponentials with no lag phase and similar amplitudes, indicating that the mutations do not impair the ability of MYO6 to bind actin and release it upon ATP-binding (Supplementary Fig. 1). The rate $k_2$ determined for ATP-binding to actin (Fig. 5f) for MYO6$^{S267E}$ was 65 mM$^{-1}$ s$^{-1}$, agreeing closely with the single-molecule data above. MYO6$^{WT}$ and MYO6$^{S267A}$, however, bound ATP at a lower rate (19.2 mM$^{-1}$ s$^{-1}$), consistent with previous data from solution kinetic studies in the presence or absence of Ca$^{2+}$[7,23] when differences in construct lengths and experimental salt conditions are taken into account (Supplementary Table 1). The slower rate $k_2$ compared with

**a**

Myosin (M)
Actin (A)

$$A.M.ADP.Pi \rightarrow A.M.ADP \overset{k_1}{\rightleftharpoons} A.M$$

$$\uparrow\downarrow \qquad\qquad \uparrow\downarrow \; k_2$$

$$A + M.ADP.Pi \rightleftharpoons A + M.ATP$$

**b**

| *Kinetics* | | [ATP] | ADP release | | ATP binding | | ATP binding | *N* | *R^2* |
|---|---|---|---|---|---|---|---|---|---|
| *Single molecule mechanics (SMM)* | | | *k1* | StdErr | *k2* | StdErr | 2nd order rate | | |
| | | [µM] | [s-1] | [s-1] | [s-1] | [s-1] | [µM-1 s-1] | | |
| S267A | | 10 | 5.30 | 0.15 | 0.52 | 0.02 | 0.052 | | 0.992 |
| | | 100 | 4.10 | 0.08 | 4.10 | 0.09 | 0.041 | 393 | 0.999 |
| | | | *mean ± SD* | | | | *mean ± SD* | | |
| | | | 4.70 ± 0.6 | | | | 0.046 ± 0.005 | | |
| | | | | | | | | | |
| S267E | | 10 | 20.50 | 2.50 | 0.58 | 0.01 | 0.059 | | 0.994 |
| | | 100 | 18.30 | 1.85 | 5.11 | 0.35 | 0.051 | 675 | 0.997 |
| | | | *mean ± SD* | | | | *mean ± SD* | | |
| | | | 19.40 ± 1.56 | | | | 0.055 ± 0.005 | | |
| *Ensemble average, displacement* | | | | | | | | | |
| S267E | | 100 | 18.00 | 4.00 | 6.00 | 1.00 | 0.06 | 79 | 0.36 / 0.48 |
| *Ensemble average, stiffness* | | | | | | | | | |
| S267E | | 100 | 20.1 | 2.2 | 7.76 | 0.01 | | 112 | 0.82 / 0.99 |
| | *d* [nm] | | 1.8 | 0.8 | 2.77 | 0.01 | | 112 | |
| | | | | | | | | | |
| | Load [pN] | 100 | | | | | | | |
| | -0.31 | | 23.3 | 0.7 | 9.6 | 0.2 | | 16 | 0.55 / 0.87 |
| | 0.68 | | 12.5 | 0.4 | 4.9 | 0.1 | | 76 | 0.74 / 0.69 |
| | 0.8 | | 16.4 | 0.3 | 4.5 | 0.2 | | 20 | 0.66 / 0.63 |
| | | | | | | | | | |
| *Literature* | | | | | | | | | |
| | | | | | | | | | |
| WT | | SMM | 5.00 a | | | | 0.070 a | | no load |
| | | SMM | 4.46 b | | | | 0.050 b | | no load |
| | | SF | 5.40 c | | | | 0.020 c | | |
| | | SF | 6.3 d | | | | 0.023 d | | |
| | | | | | | | | | |
| *Gliding-filament assay, monomeric double mutants at S267 plus T405* | | | | | | | | | |
| | velocity | **Assay buffer plus 2mM ATP, 22oC** | | | | | | | |
| | *mean ±SD* | | | | | | | | |
| | nm s-1 | | | | | | | N (filaments) | |
| S267A plus T405A | 50 ±14 | | | | | | | 51 | |
| S267A plus T405E | 41 ±17 | | | | | | | 51 | |
| S267E plus T405A | 134 ±25 | | | | | | | 51 | |
| S267E plus T405E | 133 ±35 | | | | | | | 51 | |

single-molecule studies for MYO6$^{WT}$ and MYO6$^{S267A}$ might be explained by ADP-rebinding slowing down $k_2$ in the bulk experiment.

**MYO6+$^{S267E}$ expression perturbs cortical actin dynamics**

After we observed a significant increase in ATPase activity of MYO6$^{S267E}$ in vitro, we next examined the impact of MYO6 mutations at S267 on its function in cells. Initially, we utilized the plus end-directed MYO6 mutant, in which the unique reverse gear (insert 2), the IQ motif and the lever arm extension of MYO6 is replaced by the six-IQ-domain lever arm of MYO6 (MYO6(1-770):MYO5(763-909):MYO6(913-end))[25] (Fig. 6a). Expression of this MYO6+ construct in HeLa cells results in actin reorganization at the plasma membrane and formation of filopodia-like

**Fig. 4 | Kinetics from single molecule mechanical experiments and gliding filament assays on the monomeric MYO6 phospho-mutants.** Kinetics from single molecule mechanical experiments and gliding-filament assay on the monomeric MYO6 phospho-mutants. **a** Scheme of the chemo-mechanical ATPase cycle of a single motor head. **b** Rate constants for single molecule MYO6 interactions with F-actin measured using optical tweezers at 22 °C, single trap stiffness ktrap 0.02 pN nm⁻¹. The cumulative dwell time distributions were fitted with a double exponential function, with $f(t) = A_1 \exp(-k_1 t) + A_2 \exp(-k_2 t)$. StdErr = standard error, $N$ = number of binding events in each condition, $R^2$ = regression coefficient for the fit function. The ensemble average displacement data and

stiffness data were fitted by single exponentials. The load dependence of the rates $k_1$ and $k_2$ was obtained by fitting, using $k = k_o \exp(-Fd/k_BT)$, with ko rate at zero load, F force, d distance parameter and $k_BT$ thermal energy. All single molecule mechanical data were obtained from at least 10 different motor heads in all different conditions. For comparison, values measured here and previously for MYO6$^{WT}$ in single molecule mechanical experiments (SMM) and in bulk solution (stopped-flow, SF) were included in the 'Literature' section of the table; (**a**) Lister et al.[31], (**b**) Altman et al.[33], (**c**) DeLaCruz et al.[23], (**d**) Polypenko et al.[7]. Gliding filament assay for monomeric double mutants S267A/E plus T405A/E; mean velocity and standard deviation SD.

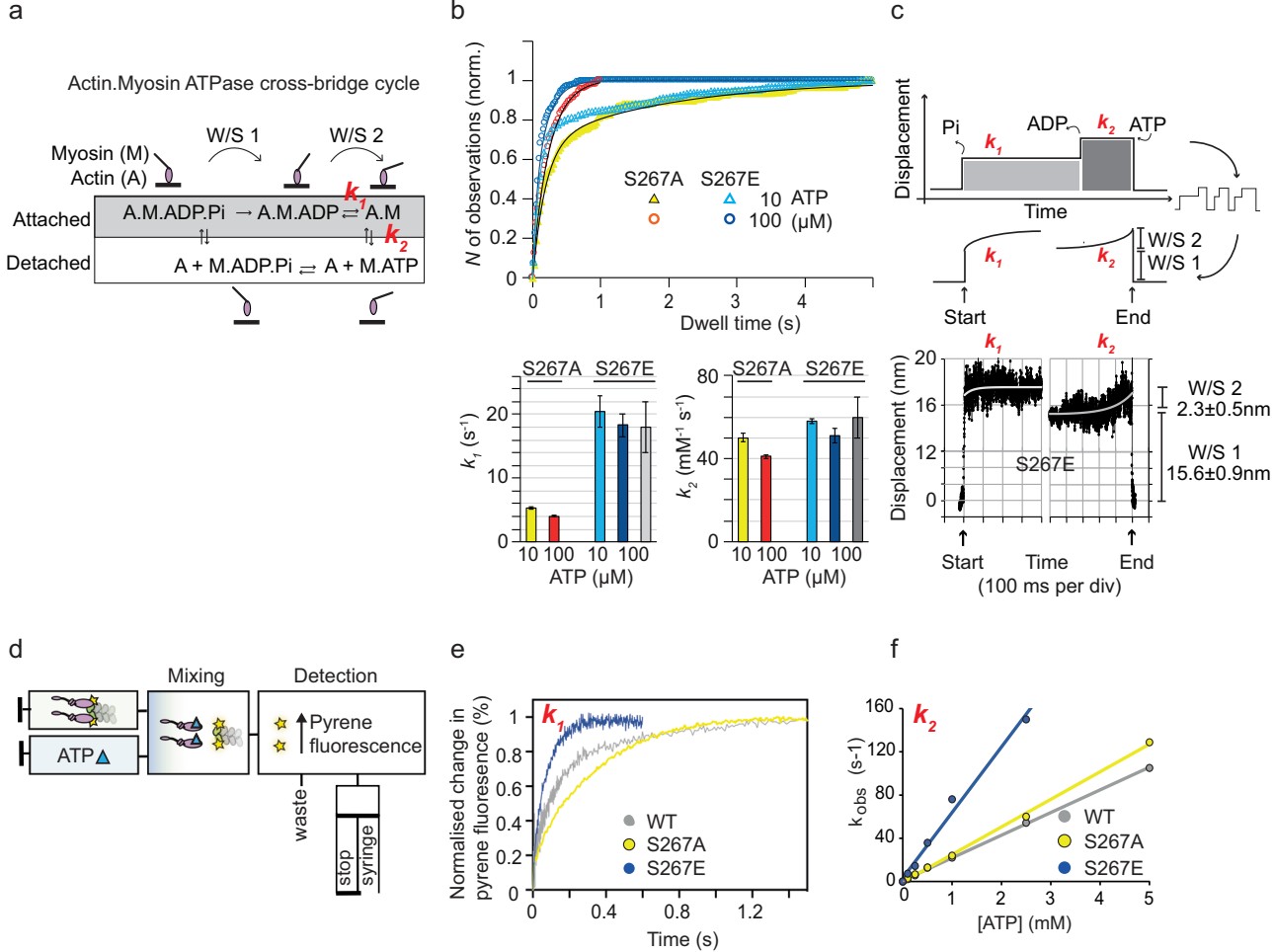

**Fig. 5 | Single-molecule mechanics and solution kinetics reveal faster ADP-release and ATP-binding rates for the phosphomimetic MYO6$^{S267E}$. a** Scheme of the actomyosin ATPase cycle. **b** Cumulative plots of the actin-attached dwell times for single MYO6$^{S267A}$ (yellow, red) and MYO6$^{S267E}$ (light and dark blue) molecules, measured with 10 μM and 100 μM ATP. Distributions can be described by two exponential components, $k_1$ (ATP-independent) and $k_2$ (ATP-dependent). Mean rates ± StdErr, see Fig. 4; data from > 3 experiments and > 3 different protein preparations. **c** Ensemble averaging of the displacement events for MYO6$^{S267E}$, with rates $k_1$ (light grey) and $k_2$ (dark grey) shown in (**b**), see also Fig. 4. **d** Scheme of

stopped flow experiments. **e** Time course of ADP displacement from a pyrene-actin.MYO6.ADP complex by ATP excess in the absence of calcium (2 mM ATP, 100 nM MYO6, 200 nM pyrene-actin, 100 μM ADP). Single exponential fits yield ADP dissociation rates ($k_1$) of 3.7 s⁻¹ (MYO6$^{WT}$, blue), 3.0 s⁻¹ (MYO6$^{S267A}$, yellow) and 16.3 s⁻¹ (MYO6$^{S267E}$, blue). **f** ATP-induced dissociation of pyrene-actin.MYO6 ($k_{obs}$) in the absence of calcium. $k_2$ values were 20.8 mM⁻¹ s⁻¹ (MYO6$^{WT}$, grey), 24.2 mM⁻¹ s⁻¹ (MYO6S$^{267A}$, yellow) and 46.4 mM⁻¹ s⁻¹ (MYO6$^{S267E}$, blue). Source data are provided as a Source Data file.

protrusions at the cell surface as previously described[25]. Induction of filopodia by MYO6+ requires motor activity, since the rigor mutant (K157R, which increases actin binding) and the D179Y mutant (that prevents processive runs on actin) in the motor domain, inhibit filopodia formation by MYO6+[25,40,41]. To determine whether MYO6 phosphorylation at S267 affected filopodia formation by the chimeric MYO6+, HeLa cells were transiently transfected with GFP-tagged MYO6+$^{WT}$, MYO6+$^{S267A}$ or MYO6+$^{S267E}$. All three MYO6+ mutants

induced the formation of filopodia at the plasma membrane and accumulated at the plus end of actin filament bundles in the tips of filopodia (Fig. 6d). We quantified the number of filopodia in cells transfected with MYO6+$^{WT}$, MYO6+$^{S267A}$ or MYO6+$^{S267E}$ and found that the number of filopodia was significantly increased in cells expressing the MYO6+$^{S267E}$ construct with an average of 120.7 ± 18.6 (mean ± SD) filopodia per cell compared with MYO6+$^{WT}$ with 65.45 ± 26.83 and MYO6+$^{S267A}$ with 51.5 ± 9.37 (Fig. 6b). The lengths of filopodia were not

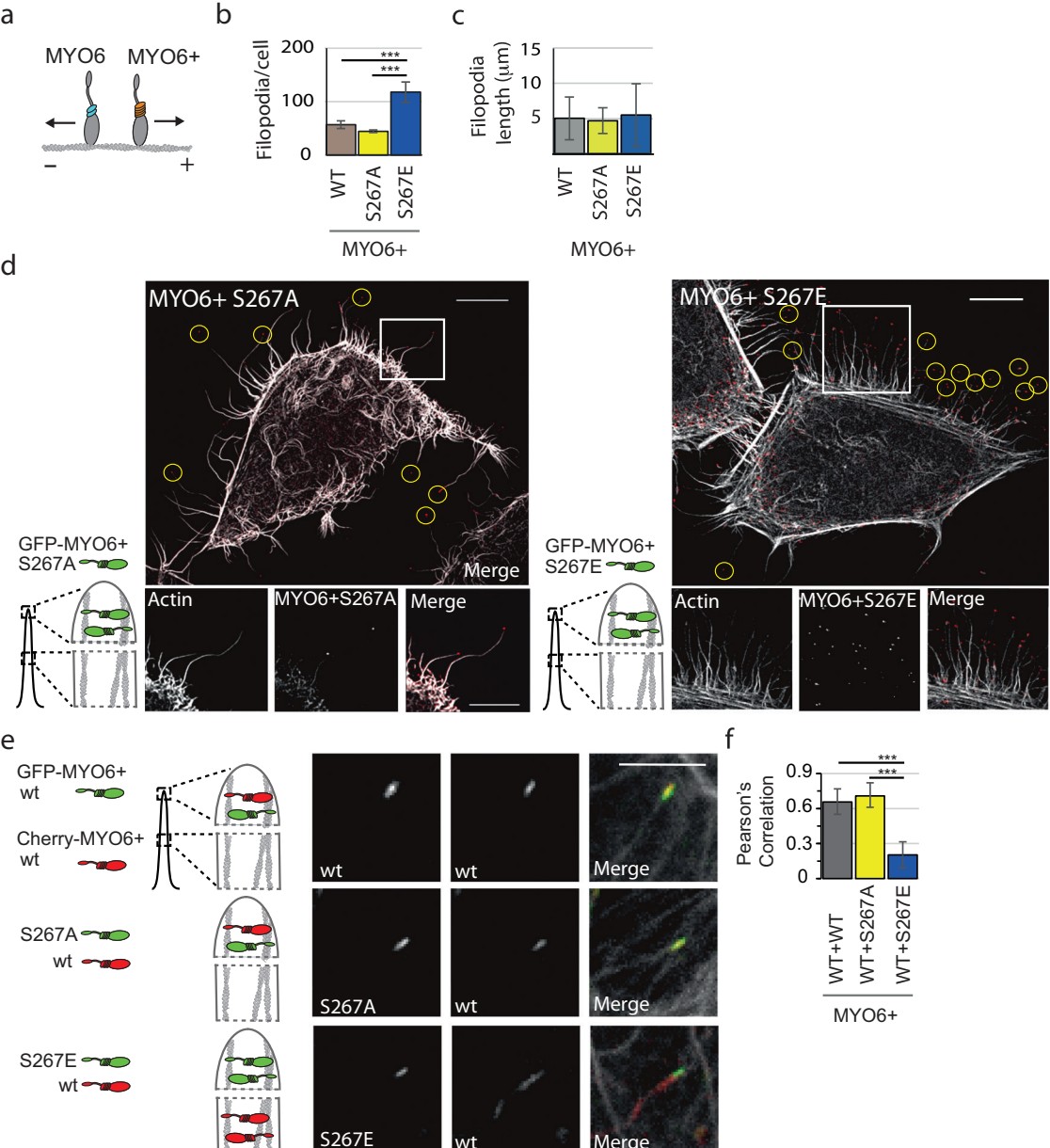

**Fig. 6 | Intracellular effects of phosphomimic MYO6^S267E. a** Scheme highlighting the design of MYO6+, the plus-end directed mutant, in which the insert-2 (reverse gear), lever-arm extension and IQ motif are replaced with six IQ motifs of MYO5. **b** Filopodia numbers per cell expressing either GFP-tagged MYO6+^WT, MYO6+^S267A or MYO6+^S267E are shown for 3 experiments. Statistical significance was determined using one-way ANOVA and post-hoc testing. ***$P < 0.01$. MYO6+^WT $n = 2786$, average 65 filopodia/cell SD ± 26.834, MYO6+^S267A $n = 1824$, average 52 filopodia/cell SD ± 9.3, MYO6+^S267E $n = 6163$, average 121 filopodia/cell SD ± 18.62. **c** Filopodia length was measured in 30–50 cells per construct in 3 experiments expressing either GFP-tagged MYO6+^WT, MYO6+^S267A or MYO6+^S267E. The lengths of filopodia were not significantly different between cells expressing MYO6+^S267A $n = 50$, average 4.72 ± 1.82 SD, MYO6+^S267E $n = 55$, average 5.5 ± 4.43 SD and MYO6+^WT $n = 56$, average 5.05 ± 3.02 SD. The lack of statistical significance was determined using a two-sided $t$-test. **d** GFP and actin localisation in HeLa cells transfected with GFP-MYO6+^S267A or GFP-MYO6+^S267E. The accumulation of either construct in filopodia

tips was observed in several independent experiments. A representative image is shown. Scale bar, 10 μm and 5 μm for enlarged images. **e** HeLa cells co-expressing mCherry-MYO6+^WT (red) and either GFP-MYO6+^WT, GFP-MYO6+^S267A or GFP-MYO6+^S267E (green) shown in high-resolution images of single filopodia. This experiment has been performed at least 3 times, representative images are shown. Schemes highlight the relative distribution of mCherry-MYO6+^WT and GFP-MYO6+^WT, GFP-MYO6+^S267A or GFP-MYO6+^S267E. Scale bar 5 μm. **f** Degree of co-localization between mCherry-MYO6+^WT and GFP-tagged MYO6+^WT, GFP-MYO6+^S267A or GFP-MYO6+^S267E was determined in 3 independent experiments from confocal images using *Pearson's* correlation coefficient with automatic *Costes* threshold. Statistical significance was determined using a two-sided $t$-test ***$P < 0.01$. mCherry-MYO6+^WT and GFP-MYO6+^WT $n = 152$, average 0.6569 SD ± 0.1054, mCherry-MYO6+^WT and GFP-MYO6+^S267A $n = 150$, average 0.7077 SD ± 0.0961, mCherry-MYO6+^WT and GFP-MYO6+ ^S267E $n = 145$, average 0.2098 SD ± 0.1123. Source data are provided as a Source Data file.

significantly different between cells expressing MYO6+^S267A (4.72 ± 1.82) or MYO6+^S267E (5.5 ± 4.43) compared with MYO6+^WT (5.05 ± 3.02) (Fig. 6c). Although these results suggest that phosphorylation of S267 enhances the ability of MYO6+ to induce filopodia at the cell surface compared with cells expressing wildtype or MYO6+^S267A, the cellular pathway involved in this process remains to be determined.

## MYO6+^S267E displaces MYO6+^WT at the tip of filopodia

We next analysed the localisation of MYO6+^WT, MYO6+^S267A and MYO6+^S267E in filopodia, to determine whether the considerable increase in velocity observed for MYO6^S267E and its decreased 'on-time' on actin filaments in vitro affected its steady-state distribution along actin filaments in vivo. To explore the effect of MYO6 + S267

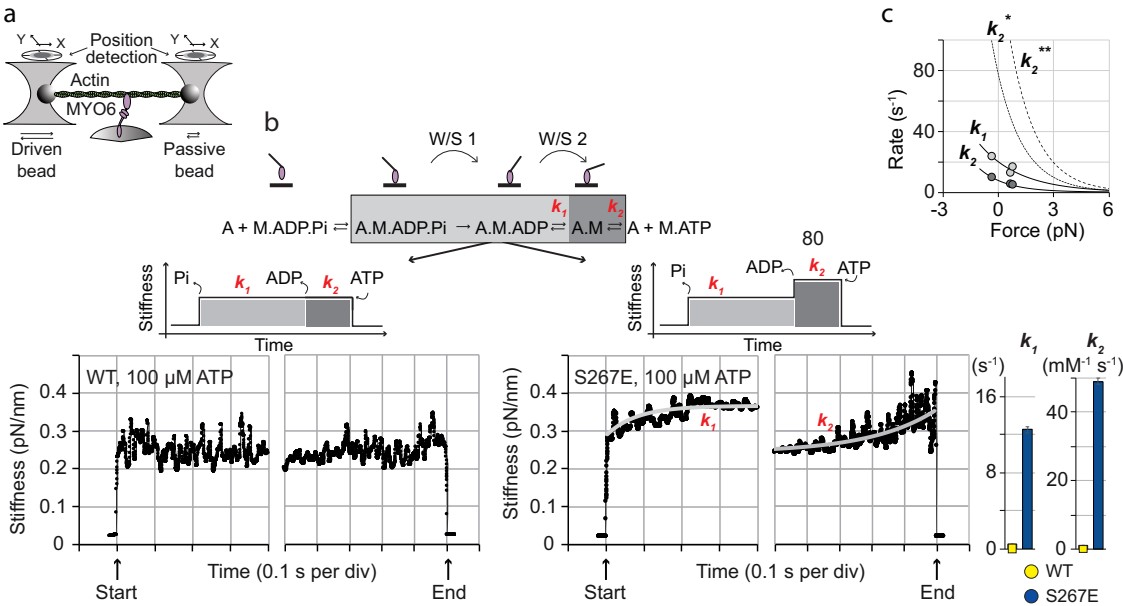

**Fig. 7 | Single molecule stiffness of acto-MYO6$^{S267E/WT}$ through the cycle.**
**a** Schematic of single molecule stiffness measurements. **b** Schematic and experimental time courses of the ensemble-averaged stiffness of the acto.MYO6 complex measured for MYO6$^{WT}$ and MYO6$^{S267E}$, see also Fig. 4. **c** Load-dependent rates $k_1$ and $k_2$ describe the change in stiffness of the acto.MYO6$^{S267E}$ complex at 100 μM ATP at different loads. Light and dark grey circles, $k_1$ and $k_2$ from ≈30 measurements at

100 μM ATP and different loads (mean ± StdErr), see also Fig. 4. Solid lines are fitted using $k = k_o \exp(-Fd/kT)$, with $k_o$ rate at zero load, $F$ force, $d$ distance parameter and $kT$ thermal energy; $d_1 = 1.8$ nm, $d_2 = 2.8$ nm. Extrapolated load-dependent rate $k_2^*$ at 1 mM ATP (dotted line) and $k_2^{**}$ at 2 mM ATP (dashed line). Source data are provided as a Source Data file.

phosphorylation on localisation in filopodia, Hela cells were co-transfected with mCherry-MYO6+$^{WT}$ and GFP-MYO6+$^{WT}$, MYO6+$^{S267A}$ or MYO6+$^{S267E}$ (Fig. 6e). As expected, co-expression of MYO6+$^{WT}$ tagged with mCherry or GFP revealed complete co-localization of the fluorophores within and at the tips of filopodia (0.66 ± 0.11 mean ± SD). Similarly, the distribution of the mCherry-MYO6+$^{WT}$ also showed significant overlap with GFP-MYO6+$^{S267A}$ (0.71 ± 0.10). Strikingly, GFP-MYO6+$^{S267E}$ displayed only minimal co-localisation with mCherry-MYO6+$^{WT}$ (0.20 ± 0.11, $P < 0.001$), appearing to displace mCherry-MYO6+$^{WT}$ from the filopodia tips and causing its redistribution along the length of the filopodia (Fig. 6e, f). Taken together, these results suggest that in our artificial cell-based assay MYO6+$^{S267E}$ and MYO6+$^{WT}$ display different motor properties which allows MYO6+$^{S267E}$ to outcompete MYO6+$^{WT}$ at growing filopodia tips.

### Single-molecule mechanics under load reveal the effect of S267 phosphorylation on MYO6 force generation

The load effect on ensembles of myosins has been investigated in gliding-filament assays slowed by variable amounts of utrophin interacting with F-actin[42,43]. To investigate the effect of S267-phosphorylation on force generation by single MYO6 motors we developed an approach using optical tweezers. We applied a 100-Hz, 100-nm peak-to-peak sinusoidal forcing function to one of the trapped beads (driven bead) during the actin-attached states and recorded the displacement of the acto-MYO6 complex using the other bead (passive bead, Fig. 7a). The applied force and induced displacement derived from the driven and passive bead positions allowed determination of the time course of the stiffness of the acto-MYO6$^{WT}$ and acto-MYO6$^{S267E}$ complexes proceeding through the cycle (Fig. 7b). By adjusting our previously introduced ensemble-averaging approach of data analysis[30,34], we found that the stiffness of acto-MYO6$^{WT}$ remained unchanged (within the noise level) at ≈0.25 pN nm$^{-1}$ from initial actin-binding to final ATP-induced detachment of the motor (Fig. 7b, WT).

For MYO6$^{S267E}$ however, the initial stiffness of ≈0.25 pN nm$^{-1}$ following actin-binding increased to ≈0.35 pN nm$^{-1}$ (Fig. 7b, S267E). Here, the transition kinetics $k_1 \approx 12.5 \pm 0.4$ s$^{-1}$ and $k_2 \approx 49 \pm 1$ mM ATP$^{-1}$ s$^{-1}$ were consistent with the rates for ADP-release and ATP-binding in the displacement measurements (Fig. 5b, c). The results show that, in contrast to MYO6$^{WT}$ and other myosin motors investigated to date, the stiffness, and thus the force generated by single phosphomimetic acto-MYO6$^{S267E}$ complexes increased strongly (≈40%) following ADP release and remained high until ATP-induced detachment from actin. Combining our quantitative stiffness measurements with the W/S (≈18 nm, Figs. 3k and 5c) enabled the stall force of (18 nm × 0.25 pN nm$^{-1}$) ≈ 4.5 pN to be determined for the monomeric MYO6$^{WT}$ motor, in contrast to ≈6.3 pN for MYO6$^{S267E}$. We next investigated whether the increased force generation following ADP-release was affected by load imposed on acto-MYO6$^{S267E}$ (Fig. 7c). Following our previous approach to determine load-dependence in single-molecule mechanical studies[32,44] we found that the effect of load on the stiffness-transition rates $k_{1/2}$ can be described by single exponentials, with $k_{1/2} = k_o \times \exp(-W/kT)$, (with $k_0$ rates at zero load on the crossbridge, $W = F \times d$, $F$ total load on the crossbridge; distance parameters $d_{1/2}$ with $d_1 \approx 1.8 \pm 0.8$ nm for the ADP-release rate $k_1$ and $d_2 \approx 2.8 \pm 0.1$ nm for the ATP-dependent rate $k_2$; $kT$ thermal energy, Fig. 4) consistent with a strain-dependent transition over an energy barrier as described by Arrhenius transition state theory. The stronger load-dependence in stiffness transition $k_2$ is consistent with previous reports on load-dependent ATP-binding for MYO6[33,45]. Extrapolating the load-dependence at 100 μM ATP to physiological millimolar ATP concentrations and assisting/resisting loads in the piconewton range showed that acto-MYO6$^{S267E}$ with increased stiffness in the nucleotide-free rigor state becomes the dominant actin-attached state of MYO6$^{S267E}$ near stall force (Fig. 7c). These results show that motor phosphorylation at a single site can lead to modified force-generating states of the acto-myosin complex generating not only a faster, but also a stronger motor.

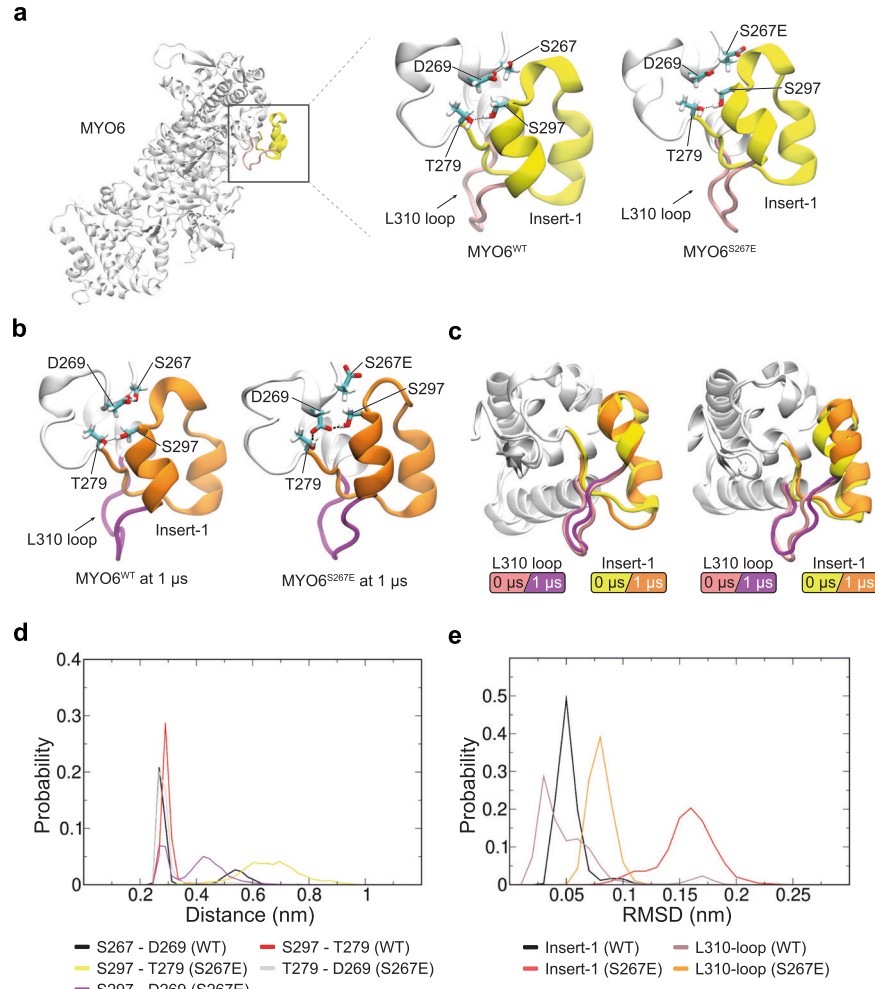

**Fig. 8 | The S267E mutation disrupts the stability of insert-1. a** MYO6 atomic model used in MD simulations. Insert-1 is highlighted in yellow. Atomic models of the insert-1 domain in WT (left inset) and S267E (right inset) mutant with key residues highlighted after energy minimization. **b** Snapshots of the insert-1 domain (orange) and 310 loop (magenta) after 1 μs for the WT (replica 2) (left) and S267E (replica 1) (right) trajectories. **c** Comparison of insert-1 and L310 loop conformations at 0 and 1 μs in WT (replica 2) (left) and S267E (replica 1) (right) mutant

trajectories obtained by superposition of residues 260–280 around the insert-1 domain and L310 loop. **d**, **e** Probability distribution plots for **d** the distance between the sidechain oxygens of the indicated pairs of residues and **e** the RMSD (root mean square deviation) of the insert-1 domain (residues 278–303) and L310 loop (residues 308–312) for the WT and S267E mutant simulations. All distance plots include data from the final 200 ns of all three replicas.

## Molecular dynamics simulations reveal impact of S267 phosphorylation on the stability of the insert-1 domain

The S267 phosphorylation site localises within the MYO6 motor domain adjacent to its unique insert-1, deletion of which is linked to increased ATP-binding[7]. To test our hypothesis that the MYO6$^{S267E}$ mutation would disrupt the interaction network between the motor domain and insert-1, we conducted all-atom molecular dynamics (MD) simulations using atomic models of both MYO6$^{WT}$ and the phosphomimetic mutant MYO6$^{S267E}$ (Fig. 8a).

While high-resolution structures of MYO6 bound to actin in the force-generating states of the cycle are not yet available, we used a high-resolution crystal structure of the nucleotide-free MYO6 from *Sus scrofa* (PDB ID: 4DBP)[41] as a template for further modelling (see *Methods*). This structure comprises the motor domain, insert-1 and a portion of the lever arm bound to one calmodulin (CaM) molecule (Fig. 8a). Each atomic model was inserted into a cubic box, solvated and subjected to independent microsecond-long MD simulations in triplicate.

Prior to the production run, in the WT system, a hydrogen bond (H-bond) is satisfied between the sidechains of S267 and D269 within the motor domain at a donor-acceptor distance of 2.6 Å, while within

insert-1 the hydroxyl groups within the sidechains of S297 and residue T279 form an H-bond at a donor-acceptor distance of 2.7 Å (Fig. 8a, left inset). In contrast, in the mutant system, the MYO6$^{S267E}$ sidechain is distant (5 Å) from the sidechain of D269 (Fig. 8a, right inset), while the S269-T279 H-bond distance is identical to the WT (Fig. 8a, right inset). To assess the impact of the MYO6$^{S267E}$ mutation, we monitored these contacts across the WT and mutant trajectories (Fig. 8a–c), summarising our findings in a distance probability distribution plot across all three simulation replicas with time-dependent analysis of the individual trajectories (Fig. 8d, Supplementary Fig. 2).

The H-bond contacts of S267-D269 and S297-T279 observed in the MYO6$^{WT}$ crystal structure were maintained overall (distance less than 0.35 nm) in the WT MD simulations (Fig. 8b–left, Supplementary Fig. 2a–c). Occasionally, loss of the S267-D269 H-bond contact occurred in all WT trajectories (Fig. 8d–black line at 0.5–0.7 nm, Supplementary Fig. 2a–c) without disrupting the H-bonding capacity between S297 and T279 (Fig. 8d–red line, Supplementary Fig. 2a–c). In contrast, in the mutant simulations, the electronegative γ-carboxyl group within the MYO6$^{S267E}$ sidechain liberates the β-carboxyl group of the D269 sidechain to initiate H-bonding interactions between the sidechain hydroxyl groups of S297 and T279, resulting in a broader

sidechain distance distribution (Fig. 8b, d−yellow line, Supplementary Fig. 2d−f).

To assess the impact of the herein established set of contacts on the insert-1 domain (residues 278−303), we calculated the RMSD for each frame in the trajectory relative to the starting model (at 0 µs) of the backbone atoms of the MYO6$^{WT}$ and MYO6$^{S267E}$ mutant insert-1 domains (Fig. 8e−black versus red, Supplementary Fig. 3a, b) and their adjacent L310 loops (Fig. 8e−brown versus orange, Supplementary Fig. 3c, d), previously associated with increased ATP/ADP exchange upon mutating the leucine residue at position 310 to alanine[7]. Superimposition of the first (0 µs) and last (1 µs) frame of the insert-1 domain shows a substantial conformational change in the MYO6$^{S267E}$ mutant insert-1 domain compared to the WT, due to the disrupted H-bonding interaction between S297 and T279 by D269 (Fig. 8b, c−right insets). The MYO6$^{S267E}$ exhibits notably higher RMSDs compared to WT (Fig. 8e, Supplementary Fig. 3a−d), underscoring the destabilising effect of the contacts observed in the MYO6$^{S267E}$ mutant within the insert-1 domain and the adjacent L310 loop.

In summary, our comparative MD simulations of WT and mutant MYO6 illustrate how the MYO6$^{S267E}$ mutation reshapes the local H-bonding network around insert-1, ultimately compromising its stability. This, in turn, induces greater flexibility in the L310-loop, potentially enhancing solvation of the catalytic site and promoting more efficient ADP/ATP exchange.

## Discussion

Here, we describe a phosphorylation event in the MYO6 motor domain and identified DYRK2 as a kinase that phosphorylates both the peptide and the full-length MYO6 at S267 in vitro. At present it is not known which physiological pathway and function requires MYO6 motor domain phosphorylation and future work will address the environmental cues regulating DYRK2 activity and its phosphorylation of MYO6 at S267. Although we have established a clear correlation between phosphorylation at S267 and changes in MYO6 kinetics and mechanics in vitro, additional phosphorylation events may regulate further aspects of MYO6 motor performance. Notably, information derived from proteomics data on the PhosphoSitePlus platform underscores that S267 and T405 represent the two most commonly detected phosphorylation sites within the MYO6 motor domain.

The unique reverse directionality is crucial for the diverse cellular roles of MYO6 that involve binding and subcellular targeting by a large variety of binding partners. Furthermore, these MYO6 cargo adaptor proteins can associate with the monomeric motor and induce dimerization, or even multimerization, allowing functional diversification and adaptation. Inevitably, the cargo binding-induced switch between a monomeric MYO6 operating as a force producer/strain sensor and a dimeric/multimeric mover/transporter is likely to require different structural and mechanical motor properties.

Specific structural and kinetic features, as well as binding partners, have been shown to regulate the mechanics of MYO6 as a monomer and dimer[10,12,45–47]. Critical differences in the gating of dimeric MYO5 compared with artificially dimerised MYO6 have shown the importance of rate limiting and load-dependent ADP (MYO5)[34,48,49] versus ATP (MYO6)[33,45,50–53] exchange for these myosins, with load affecting nucleotide exchange on lead and trail heads differently, consistent with MYO5 operating as a processive transporter, while MYO6 can act both as a transporter and an actin-based anchor[12,45,54]. Recently, regulation by phosphorylation of the motor domain at S19 was shown to structurally stabilise the rigor state and to double the in-vitro actin-gliding speed at low load for *Plasmodium* MYOA compared with a non-phosphorylatable S19A mutant[43]. An -8-fold reduction in utrophin was sufficient to arrest actin-gliding by phosphorylated MYOA compared with a mutant lacking the 19 N-terminal amino acids including S19, which strongly suggests phosphorylation to regulate gliding speed and force generation for this essential myosin in parasite invasion[43].

We addressed the effect of phosphorylation at S267 on the monomeric MYO6 mechanics and kinetics in our single molecule studies. At low (≤2 pN) load, a 4-fold increase in the ADP-release rate was observed for phosphomimetic MYO6$^{S267E}$ compared with unphosphorylated MYO6$^{WT}$. As the overall working stroke (composed of two substeps, W/S1+W/S2) remained unchanged by S267 phosphorylation, we conclude that the 4-fold increase in actin gliding-speed at saturating, millimolar ATP concentration (and low load) can be explained by accelerated, rate-limiting ADP-release. Additional phosphorylation at T405 had no effect on actin-gliding speed. Hence, translocation speed of intracellular cargo by monomeric MYO6 might be regulated by S267 phosphorylation, as was also concluded for phosphorylated *Plasmodium* MYOA$^{S19}$[43]. The 40% increase in stiffness of the acto-MYO6$^{S267E}$ complex following transition into the nucleotide-free rigor state should enable phosphorylated monomeric MYO6 to translocate intracellular cargo with a stall force increased from ~4.5 pN to ~6.3 pN per motor head. Notably, sensitivity to load characterised by the distance parameters $d_1$ - 1.8 nm (ADP-release) and $d_2$ - 2.8 nm (ATP-binding) for acto-MYO6$^{S267E}$ revealed a stronger load-sensitivity for ATP-binding compared with ADP-release, as was also concluded earlier for artificially dimerised (and unphosphorylated) MYO6[33,45]. Extrapolating the load-dependent kinetics to saturating ATP concentrations indicates that at opposing loads near the new stall force the nucleotide-free rigor state becomes the dominant actin-attached state for monomeric phosphorylated MYO6 (Fig. 7c). Therefore, if phosphorylated at S267 and dimerised, MYO6 might translocate intracellular cargo slowly, but highly processively and against higher opposing loads, with rate limiting and strongly load-dependent ATP-binding to the lead and trail heads determining translocation speed[45]. Thus, phosphorylation seems to amplify and boost the mechanical properties of MYO6 in general and turn it not only into a faster monomeric transporter at low load, but also into a slow but highly processive and strong dimeric transporter or anchor at high load. With the increased stiffness in the rigor state and stronger gating of the heads due load-dependent ATP-binding, phosphorylated dimeric MYO6 should therefore also be able to translocate cargo at higher opposing loads with increased efficiency and with longer processive run lengths. To resolve the effects of phosphorylation on physiologically dimerised MYO6 however, the stiffness measurements introduced here need to be adapted to characterise the structurally diverse, physiologically dimerised MYO6 motors. Interaction with regulating binding partners might reveal additional unexpected and regulating effects on the MYO6 monomeric, dimeric or even oligomeric mechanics.

A 100-Hz sinusoidal forcing function has been used to characterise the stiffness of other acto-myosin complexes, namely skeletal MYO2[30], smooth muscle MYO2 S1[44] and MYO5 S1[49]. No change in stiffness was detected there through the actin-attached states of the cycle. For skeletal MYO2 (immobilised on nitrocellulose) however, the ADP-release rate at low load was too fast to resolve the stiffness of ADP-bound acto-MYO2. Compared with full-length MYO6 (immobilised via a C-terminal antibody), skeletal acto-MYO2 in rigor was twice as stiff (-0.7 pN nm$^{-1}$)[30]. The calmodulin (CaM)-binding lever arm of both myosin isoforms comprises two CaM-binding motifs. In MYO2 however, the CaM-binding region is followed by a coiled-coil stabilising dimeric MYO2, while in full-length MYO6 this region is followed by a 3-helix bundle (3HB), a single alpha-helical (SAH) and a globular tail (GT) domain. Partial 3-HB unfolding and structural compliance of the minus-end directed MYO6 tail domain, suggested in MD-simulations[55], might contribute to the lower stiffness of acto-MYO6. The stiffness measured for smooth muscle MYO2 S1[44] (-0.45 pN nm$^{-1}$) and MYO5 S1[49] (-0.2 pN nm$^{-1}$) (both attached to nitrocellulose) in rigor was similar to MYO6, once differences in the CaM-binding lever arm length are taken into account. This is consistent with skeletal MYO2 being

optimised for speed and force generation, while MYO6, MYO5 and smooth muscle MYO2 fulfil less specialised, adaptable intracellular motile functions. Future structural studies will reveal which parts of the structure are responsible for stiffness and force generation as the myosin motors proceed through the actin-attached stages of the ATPase cycle.

To summarise, this phosphorylation event in the MYO6 motor domain does not indicate a simply on/off switch of the motor. Phosphorylation appears to modulate the MYO6 ATPase activity and turn it, at low load conditions, into a faster and stronger motor. Unphosphorylated MYO6 is a comparatively slow motor with a high duty ratio that spends most of its kinetic cycle attached to actin with ADP-release or ATP-binding rate-limiting transition in the ATPase cycle depending on the prevailing load and nucleotide concentrations. S267 phosphorylation induces structural rearrangements in the motor domain that lead to an increase in the accessibility of nucleotides to the ATP-binding pocket, thus enabling both faster ADP-release and ATP-binding. Under increasing cellular load, phosphorylated MYO6 seems to revert to a slow, high-duty ratio motor spending again most of its kinetic cycle attached to actin, with rate limiting nucleotide exchange regulated by the prevailing load and ATP concentrations. Furthermore, the effect of load on nucleotide exchange in the actin-attached states might affect head-coordination of phosphorylated dimeric MYO6 and increase the length of processive runs. Our results thus indicate that S267 phosphorylation induces a change in efficiency, which enables MYO6 to operate either as a faster monomeric transporter at low cellular load, or as a stronger tether and dimeric transporter moving its cargo slowly but with increased processivity against higher cellular load. Our experiments show that load-dependent transition kinetics between different states of acto-myosin stiffness are linked to specific biochemical transitions in the myosin ATPase cycle.

The in-vitro results are in agreement with results from our cell-based assay highlighting the ability of the faster phosphomimetic MYO6+$^{S267E}$ to replace the slower non-phosphorylated MYO6+ from the tips of filopodia when both are expressed simultaneously. In this scenario the higher duty ratio of WT or MYO6+$^{S267A}$ may favour attachment to actin filaments, leading to movement away from the filopodia tip driven by actin filament treadmilling, while the faster MYO6+$^{S267E}$ with lower duty ratio is able to maintain its localisation at the tip of growing actin filaments.

The capacity of MYO6 to transition between different modes of activity regulated by S267 phosphorylation/dephosphorylation may be important for its function at the apical domain of polarised epithelial cells[56–58]. In these cells a dimeric complex of MYO6 and Dab2 has been suggested to facilitate the movement of receptors and exchangers to allow clustering at the base of microvilli[59,60]. This process might require MYO6 phosphorylation to generate a fast transport mechanism. At the base of the microvilli, where clathrin-coated vesicles are clustering[61], the MYO6/Dab2 complex assumes a role in receptor uptake through clathrin-coated vesicles, which may require MYO6 to perform tethering or holding functions in its dephosphorylated state. However, at the present time the impact of MYO6 S267 phosphorylation on clathrin-mediated endocytosis is not known and further experiments are required to test the importance of DYRK2 in regulating this process. These could involve using DYRK2 KO cell lines, DYRK2 inhibitors or overexpressing DYRK2 in polarised epithelial cells.

Our results imply a molecular mechanism whereby not only the ATPase activity, but also the motor force and efficiency of chemo-mechanical energy transduction can be regulated by a single phosphorylation event, whilst the W/S of the motor is retained. With respect to the faster kinetic we demonstrate that the phosphomimetic S267E changes the local H-bonding network around insert-1 resulting in increased flexibility in these regions. In the mutant MYO6 interactions between insert-1 and the MYO6 motor domain are lost, causing repositioning and increased flexibility of the adjacent L310-loop. The

importance of insert-1 and L310 for regulating nucleotide binding has been highlighted previously using mutant MYO6 with an insert-1 deletion or L310 mutation[7]. Our study is consistent with the hypothesis of the previous report and, importantly, suggests a molecular mechanism that regulate insert-1 positioning and its role in regulating the MYO6 ATPase activity.

In summary, we have identified a phosphorylation event and the responsible kinase in the MYO6 motor domain that regulates its ATPase activity and speed of movement along actin filaments by increasing the flexibility of the MYO6-specific insert-1 and the adjacent L310-loop thereby enhancing efficient nucleotide exchange at the catalytic site. Our research highlights that a single myosin can move at different speeds and generate different forces, controlled by a single phosphorylation event, which controls the stiffness of the actin attached motor and nucleotide exchange at the ATP/ADP binding pocket.

## Methods

### Cell culture and transfection
Human cervical adenocarcinoma cells (HeLaM cells were a gift from Roger Tsien, University of California, San Diego) were cultured in RPMI-1640, retinal pigment epithelial (hTERT-RPE1 purchased from ATCC®, CRL-4000™) cells in 1:1 mixture of DMEM and F12 HAM Media and A431 cells (purchased from the European Collection of Authenticated Cell cultures, 85090402) in DMEM. All cell lines were supplemented with 10% foetal bovine serum, 2 mM *l*-glutamine, 100 U/ml penicillin and 100 μg/ml streptomycin (tissue culture media and supplements were purchased from *Sigma-Aldrich*, UK). Cells were transiently transfected 24 h prior to any experiment using FuGENE6 (Roche Diagnostics, SKU F61000) following the manufacturer's instructions.

### Immunofluorescence microscopy
Cells were grown on coverslips, washed with 1× PBS and permeabilised for 30 s with 0.02% saponin (*Sigma-Aldrich*, UK S4521) to extract cytosolic proteins and reduce the background. After fixation with 4% formaldehyde and further permeabilization with 0.2% Triton X-100, the cells were incubated with 1% BSA in PBS for 30 min, before incubating with the indicated primary antibodies, which were detected by AlexaFluor488- or AlexaFluor568-coupled secondary antibodies (*Invitrogen*, A11034 and A11031). F-actin was visualised using phalloidin coupled to either AlexaFluor568 or AlexaFluor647 (*Invitrogen*, A34055 and A22287) and cell nuclei were stained with Hoechst (Invitrogen, H3570).

### Quantification of filopodia
To quantify the number of filopodia per cell, Z-stacks were taken from at least 5 randomly selected fields, using a Zeiss *Axioimager M1* upright microscope. Maximum projections were generated in *ZEN (Zeiss)* and analysed using the *Analyse Particles* plug-in of *ImageJ* after background subtraction. Statistical significance was determined using one-way *ANOVA* and post-hoc testing to identify differences between mutant and wildtype groups. Co-localisation between mutant and wildtype MYO6 in filopodia was quantified in confocal images (*Zeiss LSM 880*) and cells of single Z-stacks analysed using the co-localization plug-in of *Volocity* (v6.3, *PerkinElmer*). Pearson's correlation coefficient was determined using an automatic threshold[62] and *P*-values and significance using Student's *t*-test.

### Constructs and antibodies
MYO6 constructs used in this study are the full-length no-insert isoform (1253 aa, UniProtKB/Swiss-Prot: Q9UM54-5, NCBI NP_001355065.1), which was used for the in vitro motility, the solution kinetic assays and the in vitro phosphorylation assays. For the inverted motility assay a zippered dimer MYO6 construct was created by adding

a C-terminal GCN4 leucine zipper sequence (MKQLEDKVEELLSKNYH-LEN EVARLKKLVGER) after amino acid 991. Single-point mutations MYO6[S267A] and MYO[S267E] were generated from full-length human MYO6 using a standard SDM protocol with the following primers (mutations underlined):

S267A-F: GAAAAACTGTACCTAAGCGCTCCTGACAGCTTCAGATAT
S267A-R: ATATCTGAAGCTGTCAGGAGCGCTTAGGTACAGTTTTTC
S267E-F: GAAAAACTGTACCTAAGCGAGCCTGACAGCTTCAGATAT
S267E-R: ATATCTGAAGCTGTCAGGCTCGCTTAGGTACAGTTTTTC.

For the MYO6 with double-point mutations at S267 and T405 four constructs were generated, including S267A plus T405A, S267A plus T405E, S267A plus T405A and S267E plus T405E. Primers used for the S267 and T405 mutants were the following:

S267A-F: ATTTGAGTGCACCAGATAATTT
S267A-R: GAAGTTTTTCTCTAATATCTTC
S267E-F: ATTTGAGTGAACCAGATAATTT
S267E-R: GAAGTTTTTCTCTAATATCTTC
T405A-F: CACCAAAGGAGCAGTTATAAAGGT
T405A-R: CCCCCTGCTGTTGTTAGCATGACT
T405E-F: CACCAAAGGAGAAGTTATAAAGGT
T405E-R: CCCCCTGCTGTTGTTAGCATGACT

Primary antibodies used in this study are mouse anti-GFP (*Abcam* ab1218), rabbit anti-dsRED (*Living Colors* 632496) and in-house MYO6 antibodies.

## Protein preparation

Actin was prepared from rabbit muscle as described elsewhere[63]. The actin was labelled with pyrene at Cys-374[35]. Briefely, F-actin was incubated at room temperature in the dark for 16 hours with a 1.5 molar excess of N-(1-pyrenyl)iodoacetamide (*Molecular Probes, Fisher Scientific*, 11599256). Unreacted label and denatured protein were removed by centrifugation at $2000 \times g$ for 1 hour. The labelled protein was then collected by centrifugation at $70000 \times g$ for 3 hours. When used at sub-micromolar concentrations, the actin was stabilised by incubation in a 1:1 mixture with phalloidin to prevent depolymerisation.

## Transient kinetics

Transient kinetic data were collected as described[64,65]. Briefly, all measurements were performed at 20 °C buffered with 20 mM MOPS, 25 mM KCl, 5 mM $MgCl_2$, 1 mM $NaN_3$ and either 2 mM EGTA or 2 mM $Ca^{2+}$-EGTA, at pH 7.0. the fluorescence signal for pyrene-labelled actin was recorded ($\lambda_{ex}$ 365 nm) and emission detected after passing through a KV389-nm cut-off filter to analyse actin-myosin interactions with ATP and ADP. Rapid-mixing experiments were completed in triplicate using a *High-Tech Scientific SF-61 DX2* stopped-flow system. Non-linear least-squares functions of transient kinetic traces were fitted initially with *Kinetic Studio, TgK Scientific*, and subsequently plotted with *OriginPro*.

## Generation of Baculovirus, protein expression and purification

MYO6 constructs were cloned into the *pFastbacHtb vector (Gibco)* and recombinant MYO6 bacmids DNA were generated by the *Bac-to-Bac* method according to the manufacturer's instructions (*Gibco*, 10584027). Baculoviral particles were generated using the *ExpiSf* expression system and *ExpiSf9* cells (*Gibco*, 15869116) according to the manufacturer's instructions. Briefly, 12.5 µg *Bacmid* DNA was mixed with *ExpiFectamine* and added to 25 ml *ExpiSF9* cells ($62.5 \times 10^6$ cells) and incubated at 27 °C for 3 days whereafter the supernatant containing the virus particles was harvested and used directly for large-scale protein purification. MYO6 (1 ml) and calmodulin virus (0.1 ml) were added to 200 ml cells and incubated until cell viability fell below 70% (approximately 3 days). Cells were pelleted at 400x RCF for 5 min and frozen ready for protein purification. Pellets were resuspended in 30 ml myosin extraction buffer (10 mM MOPS pH 7.4, 500 mM NaCl, 5 mM $MgCl_2$, 1 mM EGTA, 1 mM DTT) and sonicated for 2 min. The

extract was centrifuged at $35,000 \times g$, 30 min, 4 °C. The supernatant was combined with 1 ml Ni-NTA resin and incubated for 60 min at 4 °C. The resin was washed twice in myosin extraction buffer and twice with myosin HMM buffer (10 mM MOPS pH 7.4, 0.1 mM EGTA, 100 mM NaCl). Protein was eluted with 4 ml HMM buffer containing 150 mM imidazole. Fractions were aliquoted and snap frozen in liquid $N_2$.

Baculoviral particles of the double-point mutated MYO6 constructs were generated using the *Sf21* expression system (*Gibco*) following manufacturer's guidelines. Proteins were expressed and purified using His-affinity chromatography followed by size-exclusion chromatography[18,31].

## Bacterial protein expression and purification

GST-DYRK2 (pGEX6P1-DYRK2, DU4134) was expressed in *Escherichia coli* BL21 cells. Bacterial o/n cultures were diluted in 1 L 2× TY plus antibiotics and grown at 3 °C to an OD of 0.5 before growing at 22 °C for 50 min. After induction with 0.5 mM IPTG the cells were grown further for 5 h at 22 °C. Cells were collected by centrifugation for 30 min at $6200 \times g$ and stored subsequently at −20 °C. For protein purification, the cells were suspended in lysis buffer (20 mM Tris pH 7.5, 1 mM DTT, 1× protease inhibitor tablet, 200 mM NaCl, 1% NP40), sonicated and centrifuged at $235,000 \times g$ at 4 °C for 25 min. The supernatant was combined with gluthatione-sepharose 4B (*AP Biotech*) and incubated for 4 h at 4 °C with over-head turning. The preparation was then washed 4× with wash buffer (20 mM Tris pH 7.5, 0.1% triton) followed by a further wash with 50 mM Tris pH 8.0. DYRK2 was eluted (25 mM glutathione, 100 mM Tris-HCL pH8, 2 mM DTT) and concentrated to 1.1 mg/mL. The aliquots were flash-frozen in liquid nitrogen and stored at −80 °C.

## ADP-glow™ kinase assay

Peptides were synthesised using *Genscript* services:

MYO6 SSP EDIREKLHLSSPDNFRYLNRG MYO6 AAP EDIREKLH LAAPDNFRYLNRG

MYO6 SAP EDIREKLHLSAPDNFRYLNRG
MYO6 ASP EDIREKLHLSSPDNFRYLNRG
Dyrktide RRRFRPASPLRGPPK was purchased from *Merck* (SRP0678-1MG).

MYO6 wildtype protein was expressed and purified from Baculovirus as previously described. 5µL of the buffer master mix (40 mM Tris pH7.5, 20 mM $MgCl_2$, 50 µM DTT with or without 0.5 µg Dyrk2), was added to the different peptides or full-length no insert MYO6 protein. The time course was started by adding ATP and stopped at different time points by the addition of 5 µl *ADP-Glo* reagent (*Promega*, V6930) and mixed for a few seconds in a 384-well flat-bottom assay plate (*Corning, 3570*). After each sample incubated for at least 40 min at room temperature 10 µl *Kinase Detection Reagent* was added and incubated for 30 min at room temperature before measuring luminescence using a *BMG Plate Reader, CLARIOstar*.

## HotSpot kinase assay

In vitro profiling of 31 kinases was performed at *Reaction Biology Corporation, USA*, using the *HotSpot* assay platform (https://www.reactionbiology.com/services/kinase-assays/kinase-screening).

## In vitro *motility assay*

Procedures were adapted from gliding-filament assays[31,66,67]. Briefly, rabbit skeletal G-actin was prepared and polymerised with TRITC-phalloidin[68]. MYO6 was immobilised directly onto nitrocellulose-covered coverslips. Fluorescently labelled actin filaments were injected into the flow chamber after dilution with assay buffer (AB) containing 25 mM KCl, 4 mM $MgCl_2$, 1 mM EGTA, 1 mM dithiothreitol, and 25 mM imidazole, pH 7.4 and allowed to bind to MYO6. The assay was activated with (AB) supplemented with 2 mM ATP and an oxygen scavenger system (10−20 mM DTT and (in mg ml$^{-1}$) 0.01−0.02catalase,

0.05–0.1glucose oxidase, 1.5–3 glucose). For fluorescence imaging either TRITC-phalloidin labelled actin filaments were excited with a xenon lamp using a RFP filter set (100× magnification, *Zeiss Axio-observer* microscope); alternatively, Alexa488-phalloidin labelled actin filaments (labelling ratio 1:1) were imaged on a Nikon TI-Eclipse TIRF microscope, including an EMCCD (Andor iXon3) with a 100× immersion objective, N/A 1.49. The frame rate was 0.2–1 s$^{-1}$ for a total period of 300–600 s. Only filaments moving continuously for at least 20 frames were included in the data analysis. The gliding velocity of filaments was calculated using the analysis software *GMimPro* (www.mashanov.uk) or Fiji MtrackJ plugin. All assays were carried out at 22 °C.

### Inverted motility assay

Actin filaments were polymerised using TRITC-Phalloidin with a 1:10 ratio of biotinylated G-actin (*Cytoskeleton Inc.*) and unlabelled G-actin. Fascin was purified from *E. coli* and was used to prepare actin-fascin bundles[69]. These bundles were immobilised for 5 min on glass cover-slips coated with biotinylated BSA (1 mg ml$^{-1}$) followed by streptavidin (1 mg ml$^{-1}$). GFP-tagged dimeric wildtype and mutant MYO6 were diluted in AB-buffer containing the oxygen scavenger system (see in vitro motility assay) and supplemented with 2 mM ATP. A *Zeiss* inverted TIRF microscope with a 100× objective was used to image the fluorescent molecules at a rate of 22 frames/s for 60 s. Analysis was carried out with *FIJI* using the *MTrack2 plug-in*[70].

### Phosphoproteomics using mass spectrometry

Endogenous MYO6 was immunoprecipitated from A431 cells using polyclonal affinity purified antibodies to MYO6. Pull-downs of GFP-MYO6 were performed from stable RPE cell lines expressing GFP-MYO6. A431 and RPE cells were lysed with 1% NP-40 lysis buffer containing 50 mM Tris-HCl at pH 7.5, 150 mM NaCl, 1 mM EGTA, 5 mM ATP, 5 mM MgCl$_2$ and complete protease inhibitor cocktail (*Roche*), homogenised using a 25-g needle and clarified by centrifugation at 20,000$g$ for 15 min at 4 °C. A431 lysates were precleared with *Protein A-sepharose* beads (*Cytiva*) and incubated with MYO6 affinity purified polyclonal antibody for 2 h at 4 °C, followed by incubation with *Protein A-sepharose* for an additional hour at 4 °C. GFP-MYO6 RPE cell lysate was precleared with TBS-blocked Affi-Gel resin (*BioRad*, 1536099) before incubating for 3 h with a 10 μl bead bed of GFP-nanobody *Affi-gel* resin. A431 and RPE cells were both washed 3× with lysis buffer and twice with TBS. Proteins were eluted using SDS sample loading buffer. SDS-PAGE resolved bands were excised and the proteins reduced, alkylated and digested in-gel using trypsin. The digested peptides were collected in 0.5-ml tubes (*Protein LoBind, Eppendorf*) and dried almost to completion. Samples were re-suspended in 15 μl solvent (3% MeCN, 0.1% TFA) for immediate analysis by LC-MSMS on a *Q Exactive Plus* mass spectrometer (*Thermo Fischer Scientific*) equipped with an *EASYspray* source and coupled to a *RSLC3000 nano UPLC (Thermo Fischer Scientific)*. Peptides were fractionated using a 50-cm *C18 PepMap EASYspray* column maintained at 40 °C. A flow rate of 300 nl/min using a gradient rising from 3 to 10% solvent B in 7 min and 40% solvent B in 52 min followed by a 7-min wash in 95% solvent B. MS spectra were acquired at 70,000 resolution between m/z 400 and 1500 with MSMS spectra acquired in a DDA fashion. Data were processed in *PEAKS X Pro (Bioinformatics Solutions Inc., Canada)* using the *PEAKS* de novo search engine. Carbamidomethylation, oxidation (M) and phosphorylation (STY) were allowed as potential variable modifications with a maximum of 3 missed cleavages. Data were searched against a *Uniprot Homo Sapiens database* and a database of common contaminants.

### Optical-tweezers apparatus

A force-transducer and position-detector was custom-built around an inverted microscope (*Axiovert 200*; *Zeiss*) equipped with a *Plan-FLUAR Objective (NA 1.45, 100×; Zeiss)* and an *Optovar* 2.5× insert. The infrared laser beam (*BL-106C; SpectraPhysics*) was controlled via acousto-

optical deflectors (*N45O35-3-6.5DEG–1.06; NEOS*) and two independent tweezers were synthesized by switching between two sets of x–y-coordinates at 10 kHz[29,30]. The bead position was monitored by two four-quadrant photodetectors (*Hamamatsu*) and sampled at 5 kHz with a FPGA card (*PCI-7833R; National Instruments*) controlled by a custom *LabView (National Instruments)* programme.

### Flow-cell preparation

Experiments were performed using custom-built flow cells constructed from coverglass No1 and epoxy glue[34]. MYO6 was immobilised on the surface of the flow cells using custom-made antibodies against the MYO6 C-terminus. AB-buffer was supplemented with Mg$^{2+}$-ATP as indicated in the experiments[30–32]. Biotinylated (*Cytoskeleton*, AB07) rhodamine phalloidin-labelled actin filaments were attached at either end to ~1 μm neutravidin pre-coated polystyrene beads (*Fluo-Spheres*; F8777; *Invitrogen*), which were held in the optical tweezers, and the actin filament stretched out between the beads to achieve a connecting stiffness between actin and each bead ≥ 1.0 pN nm$^{-1}$.

### Single-molecule data collection

For displacement and dwell-time measurements a 100-nm peak-to-peak, 100-Hz sinusoidal oscillation was applied to one of the beads, whilst measuring the position of both trapped beads[30]. Recorded traces were analysed by a custom *Python* script based on the variance in bead position[71]. Displacement distributions were fitted with a single Gaussian function, with $f(x) = A/(w*sqrt(pi/2))*exp(-2*((x - xc)/w)^2$. Cumulative dwell-time distributions were fitted with a double exponential function, with $f(t) = A_1* exp(-k_1 t) + A_2* exp(-k_2 t) + y_0$.

### Ensemble-averaged single-molecule mechanics

To generate ensemble averages of single-molecule interactions, a forcing function was not applied to the trapped beads. The interaction events were synchronized at the beginning and end[34]. For the first kinetic phase, short events were extended to maintain the level reached at the end of the event (mean of last 20 data points). For the second phase, the duration of short events was extended by a 20-point mean value at the beginning of the event. The ensemble-average time course was fitted by a single exponential $f(x) = A*exp(-kx) + y_0$.

### Ensemble-averaged stiffness of acto-myosin interactions

A 100-nm peak-to-peak, 100-Hz sinusoidal oscillation was applied to one of the beads, whilst measuring the position of both trapped beads (single trap stiffness 0.02 pN nm$^{-1}$). This yields information of the applied force $F$ and the extension of the cross-bridge $Ext_{ws}$[30]. The myosin stiffness $\kappa_{MYO}$ is given by $\kappa_{MYO} = F/Ext_{ws}$. Synchronizing the beginning and end of the events yields an ensemble-averaged time course of the stiffness.

### Molecular dynamics simulations—system set up

To date, there are crystal structures for nucleotide-free WT and mutant (D179Y) MYO6 deposited in the PDB (2.4 Å (PDBID: 2BKH), 2.9 Å (PDBID: 2BKI) and 2.2 Å (PDBID: 4DBP) resolution, respectively)[5,41]. To select the most suitable atomic model for MD simulations, we assessed the geometry of all deposited atomic models in *Coot*[72] in conjunction with the available PDB validation reports. Based on this assessment, we selected the nucleotide-free MYO6-D179Y mutant from *Sus scrofa* at 2.2 Å resolution (4DBP) based on the higher quality of the model compared to the WT MYO6 (2BKH) at 2.4 Å resolution. Before we adopted the mutant as our template for further modelling, we performed structural alignment in UCSF ChimeraX[73] against the WT MYO6 model (2BKH), resulting in an RMSD of 0.37 Å over 783 residues, with most of the backbone deviation occurring within the bound calmodulin molecule. We built the WT MYO6 atomic model by replacing residues 162–190 from the MYO6-D179Y mutant structure (4DBP) with the equivalent residues from WT MYO6. Residues 352-364 and 618-642

from the motor domain and 69-82 from calmodulin were included from the WT MYO6 crystal structure (2BKI). The resulting model was visually inspected in *Coot*[72] followed by global refinement in PHENIX[74] against the 4DBP structure factors to improve the geometry of the incorporated fragments. Superimposition of our final model after energy minimisation and equilibration (see below) to the WT rigor MYO6 structure (2BKH) resulted in an RMSD of 0.52 Å across 784 residues of the motor domain. The S267E mutation was introduced into our final WT model in *Coot* by selecting the most favourable sidechain rotamer (Fig. 8a−right inset). The final WT and mutant models comprised residues D3-E815 of the MYO6 and included calmodulin (residues Q3-S147). The CHARMM-GUI web server was used to setup the system[75–77]. The atomic models of MYO6 were inserted into a cubic box of 14.6 nm³, allowing a minimum of 1 nm distance from the box edges. The system was solvated using TIP3P water. Sodium and chloride ion pairs were added to achieve a physiologically representative salt concentration of 0.15 M and additional counterions were added to neutralize the excess system charge (Supplementary Table 2).

### Molecular dynamics simulation protocol

All simulations were performed using GROMACS v2022.4 utilising the CHARMM36 additive force field algorithm suitable for protein simulations [78,79]. Energy minimisation was performed using the steepest descent method (<5000 steps) followed by a 5 ns equilibration phase (1ns in the NVT and 4 ns in the NPT ensemble) to relax the system by position-restraining all protein atoms using force constants of 400 and 40 kJ mol⁻¹ nm⁻² for the backbone and sidechain atoms, respectively. MD simulations were conducted in the NPT ensemble, each lasting 1 μs and repeated three times for every system. An integration time step of 2 fs was employed and trajectory frames were written at intervals of 100 ps. The LINCS algorithm was used to constrain all covalent bonds with hydrogen atoms[80]. Long-range electrostatics were computed by the Particle-Mesh-Ewald algorithm using a real-space cutoff of 1.2 nm[81]. Van der Waals interactions were gradually switched off between 1.0 and 1.2 nm. The Nosé–Hoover thermostat was employed to maintain the temperature at 300 K with a coupling constant of 1 ps[82,83]. The protein and solvent were coupled separately. Isotropic pressure coupling was applied at 1 bar utilising the Parrinello-Rahman barostat with a coupling constant of 5 ps and compressibility of $4.5 \times 10^{-5}$ bar⁻¹ [83,84].

### Molecular dynamics simulation analysis

VMD v1.9.4 software was used to visualise the trajectories and prepare the figures[85]. Tools within the GROMACS package v2022.4 were used to analyse the trajectories[79]. Distance plots were generated by measuring the distance between the sidechain oxygen atoms of the residues of interest. The two sidechain oxygens of aspartate were grouped as acceptor atoms and the minimum distance between them and the donor oxygen was plotted. Graphical representations were generated using the Grace plotting tool (v.5.1.22) and the GNU Image Manipulation Program (GIMP) v2.10.24.

### Reporting summary

Further information on research design is available in the Nature Portfolio Reporting Summary linked to this article.

## Data availability

All relevant data are included in the paper and/or its supplementary information files. Plasmids created and antibodies used in this study will be made available from the corresponding authors upon request. Source data are provided with this paper.

## Code availability

Custom written code is available from the corresponding authors upon request.

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

## Acknowledgements

We thank Dr. Robin Antrobus and Dr Harriet Pearson for help with the proteomics and Dr Zuzana Kadlekova for many helpful discussions. This work was supported in the Buss laboratory by a CIMR-funded studentship to JJ, a Wellcome Trust PhD studentship to TOL, the Isaac Newton Trust Cambridge, a Henry Wellcome Fellowship to CAJ, and a program grant from the Medical Research Council (MR/S007776/1). Work in the AJW laboratory is supported by the UK MRC (MR/T012412/1), Blood Cancer UK (21002) and the Rosetrees Trust (PGL22/100032). CIMR is supported by a Wellcome Trust equipment grant (108415/Z/15/Z). We like to thank Prof. Martin Zacharias (TUM) for helpful discussions and Dr. John M. Davis for his editorial assistance. Work in the Veigel laboratory by AG, MK, CB, AFM and CV was funded by DFG SFB-863, SFB-1032, Friedrich-Baur-Stiftung and the LMU-Munich.

## Author contributions

J.J. designed, performed and analysed the phosphoproteomics experiments, phylogenetic research, all kinase-related experiments, cell biology experiments, discussed the MD data, expressed and purified DYRK2, and prepared the respective figures; A.G. and M.K. built an optical tweezers apparatus, designed, performed and analysed the optical tweezers single molecule work; V.K set up, ran and analysed the MD simulation and wrote the MD results section; A.J.W. discussed and helped with the structural aspects of the study; J.J., C.B. and A.F.M. performed and analysed the in vitro motility experiments; T.O.L. identified the phosphorylation site and designed and performed the initial experiments; C.A.J. and M.G. designed and performed the stopped-flow experiments, analysed and discussed the data; J.K.-J., S.D.A, J.J., C.B., A.G., M.K. and A.F.M. expressed, purified and characterised the MYO6 proteins; C.B., N.Z. discussed results and provided structural advice; C.V. devised and analysed the optical tweezers single molecule work, prepared the relevant figures and wrote the paper; F.B. conceived the overall project, discussed results and wrote the paper. All authors discussed the experiments, commented on results and contributed to preparing the figures and writing the paper.

## Competing interests

The authors declare no competing interests.
