## [Peer Review File · Nature Communications]

Motor domain phosphorylation increases nucleotide exchange and turns MYO6 into a faster and stronger motor

Editorial Note: This manuscript has been previously reviewed at another journal that is not operating a transparent peer review scheme. This document only contains reviewer comments and rebuttal letters for versions considered at Nature Communications.Editorial Note: This manuscript has been previously reviewed at another journal that is not operating a transparent peer review scheme. This document only contains reviewer comments and rebuttal letters for versions considered at *Nature Communications*.

REVIEWER COMMENTS

Reviewer #1 (Remarks to the Author):

No further comments.

Reviewer #2 (Remarks to the Author):

Motor domain phosphorylation regulates nucleotide exchange and turns Myo6 into a faster and stronger motor.
de Jonge et al.

In this article, de Jonge et al. describe that Myo6 can be phosphorylated by DYRK2 at Ser 267 in vitro and characterize the effect of this specific phosphorylation via several in vitro assays using a phospho-mimic mutant. They show that phosphorylation leads to 4-fold increase in motor speed. They also characterize a phospho mimic mutant with single molecule optical tweezers, which presents an increase in stiffness when the motor works under load, which turns Myo6 into a stronger motor. The in vitro assays are convincing and indicate how a modification in the U50 can tune Myo6 activity. In this regard, a number of new findings are provided by this manuscript and they are of interest for the description of the role of molecular motors, in general.

Less convincing is the fact that little phosphorylated Myo6 is found in cells. It is unclear whether Myo6 phosphorylation at S267 is really important for a cancer cell, while the authors describe the specific kinase as having a role in cancer. The cell experiments described are in large part artificial (Myo6+ chimera), and they do not directly indicate the function Myo6 would play in the cellular roles examined and how phosphorylation could play a role.

Overall, the manuscript studies an important theme for motor proteins : the fact that they may be tuned by phosphorylation not only as an on/off switch but also to tune the precise motor activity they may need to perform in cells. There is no doubt on the quality of the experiments but the result for importance of this phosphorylation in cell is much less convincing compared to the change in properties found from in vitro experiments. In addition, the molecular dynamics experiments have been revised. They are now done correctly but they are restricted to a very small region of the motor and their presentation should be revised.

The discussion would gain by summarizing what was previously described in the literature for the properties and mechanism used by Myo6 to play a role as a transporter and an anchor, and what is added by this study regarding the rate limiting steps and difference in motor properties when the motor is phosphorylated at position of the HCM loop or on S267.

Overall, while the authors have found a potential site of phosphorylation of Myo6 that could tune motor properties, the mechanism by which this occurs cannot be described by the molecular dynamics the authors provide, and there is little evidence that this phosphorylation is used in cells to tune myo6 activity. The interest in the study remains in the thorough analysis of what a phospho-mimic mutation changes in the motor properties, including very elegant work on single molecule studies. It would be best for the authors to state what is strongly demonstrated in this study. Acknowledgment of the need of further investigation (and the description of the limits of the study) would be less confusing for readers than a text with too many hypothesis without any demonstrated evidence that this would be linked to phosphorylation, in particular since the ability of Myo6 to function both as a transporter and an anchor may not require this phosphorylation.

Main comments regarding the rebuttal

Regarding comments of reviewer 1 and 2 (comment 10-11) regarding molecular dynamics simulations :

The answers to rebuttal of reviewer 1 seem mainly ignored, leaving still unsatisfaction in the work done on molecular dynamic simulations and how they can provide answers regarding to how the function and stiffness of the motor differs upon phosphorylation.

The modelling studies are limited to the change in dynamics near the phospho-mimic mutation. The changes identified cannot explain how this influences the stiffness of the motor. They may not be relevant to indicate how the motor operates under load. They can explain the role the phosphorylation can have for the release of nucleotide. However, the presentation of these results should be improved, as it is currently how this part of the motor rearranges during the simulation.

1) Molecular dynamics : correct the way the two simulations are compared – line 381 : distance is indicated but it must be compared to what it was at first, otherwise it does not indicate the movement that occurs. This part is badly written to evaluate what occurs during the simulation : it is unclear whether large or small changes occur. In fact, a phosphoserine could also be mimicked with at least one simulation to be sure what is observed is similar to what occurs with S267E, to evaluate whether the phospho mutant behaves like a phospho-serine. See also the questions raised about how these simulations were done in minor comments below. It would be important to describe and illustrate better what was done.

Regarding comments of reviewer 2 :

Lack of evidence / comments on the cellular role that could play a phosphorylation on S267 (comment 1).

- The cell biology experiments described in the manuscript are diverse and inconclusive for describing whether phosphorylation on S267 would be critical in a particular cellular process.

It is recommended that the authors state this fact and suggest further experiments that could potentially lead to investigate the role of S267 phosphorylation. Instead of summarizing with the current literature what should be the conclusion from cell assays described in the manuscript, the authors discuss potential hypothetical roles of Myo6 in other cellular processes. It is important to indicate what is in favour of or what should be done to test that transient phosphorylation could bring an advantage to (1) a particular cell process upon external stimuli, (2) Salmonella invasion. The third example chosen is artificial (Myo6+) and thus does not help fully to validate a role of this phosphorylation site for Myo6 in a biological process.

The role Myo6 may play in Salmonella invasion is not described and thus why phosphorylation would increase its efficiency is not linked to the properties found in vitro. It is unclear that phosphorylation is induced by Salmonella during invasion, and the results found with the phospho-mimic S267A and S276E mutants are not strikingly different. The Myo6+ assay is also an 'artificial' as they do not probe a role of Myo6 performed in a regular cell, although they demonstrate an ability of the mutation to change the way Myo6+ can promote filopodia initiation. Thus there is less convincing data in this manuscript regarding the role of this phosphorylation for a particular role of Myo6 in cells.

- In discussion, instead of summarizing the unresolved questions from the results, the authors describe in general terms possible sequential roles of Myo6 in microvilli but no reference is provided. Is it clear that no endocytosis occurs at the tip of microvilli and that this is only found at the base? A statement indicating what type of experiments could be an approach to test the role of phosphorylation in cell processes in future studies would be important. There is currently no clear evidence for a kinase that would be efficient for the phosphorylation of Ser267 or a consequence in the lack of this kinase for different cellular roles of Myo6. It would be important that the authors acknowledge that there is currently no evidence for the need of Myo6 phosphorylation to perform these functions.

- In the view of this reviewer, for consistency, it is also important to clarify the hypotheses regarding the role in cancer (as the manuscript describes that the kinase involved in Ser267 is overexpressed in cancer).

- Currently, the reader may wonder whether the study of S267E describes an 'interesting' mutant indicating that Myo6 properties could differ by a mutation in this place, rather than a study of the role of phosphorylation of myo6 required for distinct specific functions / cellular roles of myo6 in distinct places/life time of the cell. The authors should be more clear and modest in results/discussion for what they can conclude from their cellular assays. There is a tendency to 'sell' the potential role of an interesting potential modulation without much evidence for this to be really occurring in cells.

Major detailed comments regarding this point :

2) The assays at a cellular level are not convincingly indicating that Myo6 uses phosphorylation to

perform a particular cellular function. The fact that only 1-2 % of Myo6 is phosphorylated, whatever the stimulation with EGF or PMA makes it unclear whether the phosphorylation is really important for cancer cells, unlike what is proposed/indicated in the manuscript when the link of DYRK2 with cancer is mentioned.

The current manuscript is in fact having only limited data for indicating that the motor gets phosphorylated in cells, and thus needs to study a phospho mutant, which is not the best way to indicate why a cellular function would require a distinct cellular function controlled by phosphorylation. Data from a mutant indicating that the lack of phosphorylation leads to the apparition of a phenotype in cell would be more convincing.

Additional evidence for the role of the phosphorylation in cancer cell survival, proliferation or migration would be required to keep the text that mentions cancer roles of DYRK2.

In sum, there is not a clear demonstration of how Myo6 is regulated by phosphorylation and which function would be affected if the residue would be unable to be phosphorylated (S267A mutant).

3) In addition, the phosphorylation performed with DYRK2 in vitro from FL Myo6 or fragment of Myo6 would be expected to be efficient if Myo6 is a substrate of this kinase, which is not the case. This is also demonstrating the poor efficiency of the phosphorylation event.

Overall, there is a doubt that this phosphorylation plays a role in cells and the authors mainly focus on studying phospho-mimic mutants, which limits the study.

4) Line 292-294: it is not clear whether the chimera MYO6+ works under no load in these assays and whether this conclusion can be suggested. The filopodia assays indicate that the phosphorylation-mimic mutant increases the number of filopodia but this is difficult to reconcile with the role Myo6+ has since the chimera is not described and it is thus difficult to understand how this artificial motor promotes filopodia and why phosphorylation might impact its localisation and function.

5) Salmonella entry assay and Figure 5i. The assay does not seem to be robust as the data points show large discrepancy, among different assays performed with the mutant S267A. Moreover, the role of Myo6 in Salmonella entry should be better summarized based on previous studies/hypothesis. Is it possible to propose why the phosphorylation of the motor would influence the entry of Salmonella ?

6) It is important to explain specifically how stiffness is measured and how it compares to other motors as the authors indicate that the phosphorylation event is modulating this parameter. It might be a better choice in discussion to provide a clear statement of the changes in the properties of the motor that would result from phosphorylation and if this would make the motor more performant on all cellular tasks / or more performant only on some of them.

7) Discussion : it is important to summarize the kinetics and the model for describing Myo6 processivity and duty ratio that were previously proposed by other groups. In particular, how the assays performed here compared to WT Myo6 studies in the Sweeney lab would be informative. It seems that ATP binding was also previously described to be the rate limiting step and the model allowing control of gating between heads for Myo6 was proposed to differ from that of Myo5. This should be discussed here. Also what was measured as stall force for Myo6 or Myo5 should be added line 433-434. See Sweeney et al Embo J 2007. Taking in consideration papers of Myo6 and Myo5, lines 446-448 should be revised.

8) Discussion of different hypotheses on the role phosphorylation might have in endocytosis seems to be pure speculation. It would be important to be more precise on what is known about endocytosis in microvilli and to provide references. There is no discussion on how phosphorylation could be triggered, and by which kinase. Instead of hypothetical roles of myo6 in cells for which no kinase is important for regulation, it would be of interest to propose follow up experiments that could describe what are the cellular processes that depend on kinases and Myo6 that could be

9) be more precise on the mechanism on different cellular roles myo6 may have in cells that do not

have any evidence about the importance of a phosphorylation event of Myo6 or a mutation on Ser267 as having a phenotype.

a. Lines 455-458 and the rest of the discussion : there is no strong ground in linking the different roles of Myo6 with the diverse cellular processes

Additional articles that previously indicated the role of phosphorylation for regulation of speed and processivity (Comment 1)

- The authors do not state correctly the findings of the Robert-Paganin et al. 2019 article. They only mention the methods.

o Lines 316-317 : the sentence used to indicate the results of Robert-Paganin et al. 2019 is not appropriate. Only the methods are described and there is no acknowledgement of the results that this study and already described the possibility to tune a motor for different roles (fast motility / load resistance) depending on phosphorylation.

o Lines 493-496 : Previous studies on PfMyoA have indicated that a single phosphorylation site regulates the duty ratio of the motor and that this phosphorylation for the invasion of the parasite. This study (Robert Paganin et al) should be cited as a publication establishing the importance of phosphorylation in controlling motor function. And 'for the first time' should be omitted as this PfMyoA study as well as those previously mentioned for phosphorylation of the HCM loop in the control of other myosins should be mentioned in discussion.

Additional phosphorylation sites (comments 2,3, 4)

- The authors have corrected errors and improved the manuscript and the changes are satisfactory.
- The authors do not reveal clearly if their analysis of the proteome lead to the identification of other potential Myo6 phosphorylation sites. The authors could cite their mass spec results, without having to identify the consequences of each of these mutations. However, it is of interest to depict whether other phosphorylation sites have been previously described elsewhere in the Myo6 motor and if these sites were also phosphorylated in the experiment depending on external cellular signals, in particular since little amount of phosphorylation is found on S267.

Constructs used in these studies (comment 5)

Unlike stated in the rebuttal, the authors do not describe clearly which Myo6 constructs they are purifying and using. They mention the isoform but not the exact length of the construct used. In addition a clear description of the Myo6+ construct should be provided.

For example, the constructs used for the transient kinetic assays should be precised in the main text (and in methods). Currently references are mentioned in methods without clearly establishing the constructs used for these assays. In fact, it is important to mention the construct used clearly for all the in vitro / in cell assays.

For example : Line 258-259 : whether the tail is part of the construct is not mentioned – the lever arm extension is not introduced / defined, whether the constructs can dimerize is not clearly mentioned.

Ca²⁺ and phosphorylation (comment 7)

- Without a clear description of what is known about motility in presence of Ca²⁺ and cargo, it is unclear that what is described in rebuttal is full speculation. The explanations are unclear and speculative. The authors are not linking directly the relation that would occur between amount of Ca²⁺ and phosphorylation and should state what they propose more clearly.

Myo6+ and filopodia results (comment 8)

- The authors have taken into account the previous modifications requested. The fact that the S267E mutant and WT would not lead to mix motors is still unclear that the authors did not explain. It would be of interest to know what are the differences between the constructs, besides the mutation.

Invasion assays (comment 9)

- The authors have taken into account the previous modifications requested. The comment about

figure 4g was missing not : "The data presented in 4g does not seem to show ... So this comment was confusing as written.

- However, the fact that the difference is not so strong between the A and E mutant is not in favour of indicating a strong phenotype depending on Myo6 phosphorylation, see comment 1, which does not lead to a good assay to support the significance of this potential phosphorylation site.

Minor comments

- The title should not only indicate that phosphorylation regulates nucleotide exchange, but also indicate whether this exchange is increased/decreased
- Please indicate in methods and in the legend of figures which constructs allowed to perform the different assays. This is not clear for example for the kinetic studies. Thus, it is difficult to interpret the data, in particular the role of Ca²⁺ in modulating ADP release.
- Line 72-73 : one recent study published on Myo6 specifically address the distinct role of cargo-adaptor proteins in motor activation and dimerization and should be added here : Canon et al. 2023.
- Line 81-82 : similarly, this question was addressed by Canon et al and should be mentioned in introduction or discussion.
- Line 127 : why use an alphafold structure of Myo6 to show where S267 since there is actual data of the motor : high resolution structures of Myo6 ?
- Line 132 : mention to a figure is missing to state that the levels of phosphorylation is 1-2%
- Line 155 : the graph 2f is a bit misleading since it only show the top of the graph. It would be important to represent the phosphorylation also as done in figure 1f.
- Line 195 : mutant is missing after MYO6S267A-mutant
- Line 195 : add nm for the working stroke
- Line 209 / 246 / 248 and elsewhere – the rates are mentioned as mM-1 rather than μM
- Line 214 : W/S 2 is not a state but a transition
- Line 217 : 34 is not the right publication, structure of Myo6
- Lines 236-240 : Please introduce why testing the role of Ca²⁺ in transient kinetics is of interest.
- Lines 254-255 or in methods : describe precisely the chimera Myo6+ (which residues of Myo6 and Myo5 are included in the construct). It is not easy to find the information in the references mentioned.
- Line 299 : 'recently' please provide a reference
- Line 316-317 : the sentence is backwards. It is unclear why the authors introduce the reference 40 and 41 only to cite the reference as a method, rather than for citing the results describing that PfMyoA motility modulated by phosphorylation. This was asked in the previous review and the rebuttal indicate that it was corrected but this is not the case.
- Line 354 : the experiment performed is a phospho-mimic (mutation) rather than a phosphorylation and it might be safer to use "phosphorylation at a single site may lead to ..."
- Line 365-366 : the PDB chosen is that of a mutant form of Myo6 - Please add reference here and comment in methods where the main chain of the starting model of the simulation after relaxation differs from that of the WT rigor myosin. The use of the mutant PDB 4DBP as a starting model is not justified and it seems odd to use structure factors to restrain the structure. The method used by the authors and the simulation results should be clarified. It would be important to show also that the minimization has reached convergence and describe better what are the consequences seen in the flexibility of the region near the phosphorylation and elsewhere in the motor.
- Line 381 : 1 nm apart : please indicate the atoms from which this measure is taken and how it differs in WT and S267E
- Lines 411-414 : sentences are unclear.what are the additional phosphorylation events referred to here ? Only these two are mentioned – are there any other found in PhosphoSitePlus ?
- Line 460 Note that a recent publication indicated that GIPC does not induce multimerization but dimerization of Myo6 (Canon et al Nat Comm 2023).

Reviewer #3 (Remarks to the Author):

As discussed with the editor I am only able to make limited comments on the experimental methods that use optical tweezers. The three bead assay as used here was pioneered by some of the authors. As in the previous submission I find these methods to be well and clearly described, with good quality figures to illustrate both methods and results.

The authors have corrected the repeated citations of the previous versions.

POINT-BY-POINT RESPONSE TO THE REVIEWERS' COMMENTS

Reviewer #1 (Remarks to the Author):

No further comments.

Reviewer #2 (Remarks to the Author):

Motor domain phosphorylation regulates nucleotide exchange and turns Myo6 into a faster and stronger motor.
de Jonge et al.

In this article, de Jonge et al. describe that Myo6 can be phosphorylated by DYRK2 at Ser 267 in vitro and characterize the effect of this specific phosphorylation via several in vitro assays using a phospho-mimic mutant. They show that phosphorylation leads to 4-fold increase in motor speed. They also characterize a phospho mimic mutant with single molecule optical tweezers, which presents an increase in stiffness when the motor works under load, which turns Myo6 into a stronger motor. The in vitro assays are convincing and indicate how a modification in the U50 can tune Myo6 activity.

In this regard, a number of new findings are provided by this manuscript and they are of interest for the description of the role of molecular motors, in general.

“Less convincing is the fact that little phosphorylated Myo6 is found in cells.”

We agree that we predominantly observe low-level phosphorylation of MYO6 in retinal pigment epithelium (RPE) cells. Nonetheless, phosphorylation data of MYO6 from PhosphoSitePlus, an open-access systems biology resource, does confirm the presence of phosphorylation at S267 and T405 within the motor domain of MYO6. Phosphorylation at S267 has been documented in multiple high-throughput studies, which often used cancer cell lines as their experimental models. (Please see screenshots of the PhosphoSitePlus website below. S267 as well as T405 are the most frequently identified phosphorylation site in the MYO6 motor domain.

“It is unclear whether Myo6 phosphorylation at S267 is really important for a cancer cell, while the authors describe the specific kinase as having a role in cancer. “

Interestingly, in most of the high throughput data sets on phosphosite plus, which identified MYO6 S267 phosphorylation, cancer cells such as breast cancer, lung cancer, non-small lung cancer, non-small cell lung adenocarcinoma have been used. Furthermore, it is well documented that MYO6 is consistently upregulated in many different cancers linked to cancer progression and outcome prognosis. In addition, there is a strong link between DYRK2 in cancer:

Correa-Saez, A. et al. Updating dual-specificity tyrosine-phosphorylation-regulated kinase 2 (DYRK2): molecular basis, functions and role in diseases. *Cell Mol Life Sci* **77**, 4747-4763 (2020).

Tandon, V., de la Vega, L. & Banerjee, S. Emerging roles of DYRK2 in cancer. *J Biol Chem* **296**, 100233 (2021).

Therefore, we did not feel that including a potential link between MYO6 and DYRK2 was too farfetched.

However, we now have deleted all the passages from the manuscript that discuss a potential link between DYRK2 and cancer in the abstract, introduction, results and discussion.

“The cell experiments described are in large part artificial (Myo6+ chimera), and they do not directly indicate the function Myo6 would play in the cellular roles examined and how phosphorylation could play a role. “

We now describe the experiment using the MYO6+ chimera as a “cell-based assay” and we do not draw any conclusion which physiological role we are examining.

“Overall, the manuscript studies an important theme for motor proteins: the fact that they may be tuned by phosphorylation not only as an on/off switch but also to tune the precise motor activity they may need to perform in cells. There is no doubt on the quality of the experiments but the result for importance of this phosphorylation in cell is much less convincing compared to the change in properties found from in vitro experiments. “

To determine the exact spatial and temporal importance of MYO6 S267 phosphorylation in cells has been challenging, as the signalling pathways that regulate phosphorylation are not known. We therefore have toned down our conclusions on the importance of phosphorylation in cells.

“In addition, the molecular dynamics experiments have been revised. They are now done correctly but they are restricted to a very small region of the motor and their presentation should be revised.”

We are pleased to learn that reviewer 2 now approves our molecular dynamics experiments and states that they have been done correctly. In our MYO6 motor-CaM system we did not observe major conformational changes away from the insert-1 domain comparing the WT and S267E phosphomimetic trajectories. We therefore

focused on assessing changes linked to S267 phosphorylation and the local H-bonding network around S267 and insert-1.

“The discussion would gain by summarizing what was previously described in the literature for the properties and mechanism used by Myo6 to play a role as a transporter and an anchor, and what is added by this study regarding the rate limiting steps and difference in motor properties when the motor is phosphorylated at position of the HCM loop or on S267.”

We have now included a more detailed discussion of previous in-vitro work on the chemo-mechanical properties of monomeric and dimeric MYO6, also in comparison with dimeric MYO5. We have also expanded the discussion on the chemo-mechanical effects of MYO6 phosphorylation when acting as a monomer or dimer (line 420-427).

“Overall, while the authors have found a potential site of phosphorylation of Myo6 that could tune motor properties, the mechanism by which this occurs cannot be described by the molecular dynamics the authors provide, and there is little evidence that this phosphorylation is used in cells to tune myo6 activity. The interest in the study remains in the thorough analysis of what a phospho-mimic mutation changes in the motor properties, including very elegant work on single molecule studies. It would be best for the authors to state what is strongly demonstrated in this study. “

Please see our detailed comments below to these points. We have now toned down our conclusions on the cellular consequences of MYO6 S267 phosphorylation. As described above we have extended the discussion on the effects of phosphorylation on the molecular mechanics of MYO6 and compared motor stiffness measurements in this study with previous work on MYO2 and MYO5 (line 435-515).

“Acknowledgment of the need of further investigation (and the description of the limits of the study) would be less confusing for readers than a text with too many hypotheses without any demonstrated evidence that this would be linked to phosphorylation, in particular since the ability of Myo6 to function both as a transporter and an anchor may not require this phosphorylation. “

In the discussion we have removed the paragraph speculating on the impact on MYO6 S267 phosphorylation on the function of MYO6 on endosomes linked to different cargo adaptor proteins such as GIPC or TOM1/2. Furthermore, we have modified our example on two potential roles of the MYO6/Dab2 adaptor complex in clathrin mediated endocytosis in polarised epithelial cells and now state the limitations of this example: “However, at the present time the impact of MYO6 S267 phosphorylation on clathrin-mediated endocytosis is not known and further experiments are required to test the importance of DYRK2 in regulating this process. These could involve using DYRK2 KO cell lines, DYRK2 inhibitors or overexpressing DYRK2 in polarised epithelial cells.”

We have now also acknowledged the need of further in-vitro investigation of the chemo-mechanical properties of physiologically dimerised MYO6 in the presence of regulating binding partners (line 523-528).

“Main comments regarding the rebuttal

Regarding comments of reviewer 1 and 2 (comment 10-11) regarding molecular dynamics simulations:

The answers to rebuttal of reviewer 1 seem mainly ignored, leaving still unsatisfaction in the work done on molecular dynamic simulations and how they can

provide answers regarding to how the function and stiffness of the motor differs upon phosphorylation.”

We took great care to answer the questions of reviewer 1 and he has “no further comments”.

“The modelling studies are limited to the change in dynamics near the phospho-mimic mutation. The changes identified cannot explain how this influences the stiffness of the motor. They may not be relevant to indicate how the motor operate under load. “

We thank the reviewer for this comment. However, the current lack of high-resolution atomic models for the full length MYO6 bound to actin is a limiting factor for meaningful MD simulations. Measuring stiffness of the motor in absence of actin and cargo would not provide results that can be compared to the stiffness measured experimentally in this work.

As we stated in the initial rebuttal letter and in the results section, the MD simulation work was completely revised with the aim of explaining the role that phosphorylation and the phosphomimetic mutation has on the release of nucleotide.

“They can explain the role the phosphorylation can have for the release of nucleotide. However, the presentation of these results should be improved, as it is currently how this part of the motor rearranges during the simulation.

1) Molecular dynamics: correct the way the two simulations are compared – line 381: distance is indicated but it must be compared to what it was at first, otherwise it does not indicate the movement that occurs. This part is badly written to evaluate what occurs during the simulation : it is unclear whether large or small changes occur. In fact, a phosphoserine could also be mimicked with at least one simulation to be sure what is observed is similar to what occurs with S267E, to evaluate whether the phospho mutant behaves like a phospho-serine. See also the questions raised about how these simulations were done in minor comments below. It would be important to describe and illustrate better what was done.”

We thank the reviewer for the comment. We have now modified the text and added two new panels in Figure 7c showing a superimposition of the first and last frames of the insert-1 domain in the WT (replica 1) and S267E (replica 1) MD simulations. We have also stated the distances in the text.

In the first round of the reviewing process, reviewer 1 was opposed to studying the effect of the phospho serine mutant in the MD simulations because the experimental work was done using the S267E mutation. However, as the reviewer requested, we show here in the rebuttal letter the results of a 1 μ s simulation using the phosphoserine modification. We observed a similar behaviour to the phosphomimetic S267E mutant with the sidechain of D269 switching to contact the sidechains of T279 and S297 (see supporting figure below). We conclude that although the disruption of the T279-S297 hydrogen bond was less prominent in this single simulation run compared to the S267E simulations, the electronegative phospho-S267 indeed drives D269 to contact T279 and S297, disrupting the stability of the insert-1 domain.

Supporting figure for the PhosphoS267 simulation. Left, timeline showing that consistent with the data for the S267E mutant, residue D269 intermittently disrupts the H-bond contacts between the sidechains of S297-T279 throughout the trajectory. This is not observed in the WT (see Extended Data Figure 2a-c). Right, Residues 260 to 280 of the motor domain are superimposed at 0 μs and 0.94 μs . Insert-1 domain and the L310 loop are highlighted at 0 μs (yellow and pink) and 0.94 μs (orange and magenta). Key residues at 0.94 μs are represented as sticks.

“Regarding comments of reviewer 2:

Lack of evidence / comments on the cellular role that could play a phosphorylation on S267 (comment 1).

• The cell biology experiments described in the manuscript are diverse and inconclusive for describing. Instead of summarizing with the current literature what should be the conclusion from cell assays described in the manuscript, the authors discuss potential hypothetical roles of Myo6 in other cellular processes. It is important to indicate what is in favour of or what should be done to test that transient phosphorylation could bring an advantage to (1) a particular cell process upon external stimuli, (2) Salmonella invasion. The third example chosen is artificial (Myo6+) and thus does not help fully to validate a role of this phosphorylation site for Myo6 in a biological process.”

In response to these comments of reviewer 2, we have now suggested additional experiments in our discussion that could be performed to further validate the role of MYO6 S267 phosphorylation in different cellular processes: (1) “further experiments are required to test the importance of DYRK2 in regulating this process. These could involve using DYRK2 KO cell lines, DYRK2 inhibitors or overexpressing DYRK2 in polarised epithelial cells.”

(2) In line with suggested changes we have removed our experiments using Salmonella invasion.

(3) We agree that using MYO6+ does not test the role of S267 phosphorylation in a biological process, but generates a cell-based assay system that allows to compare impact of S267E on cellular distribution relative to S267A. We now clearly state that

the MYO6+ is a cell-based assay and do not claim that it simulates a physiological pathway.

“The role Myo6 may play in Salmonella invasion is not described and thus why phosphorylation would increase its efficiency is not linked to the properties found in vitro. It is unclear that phosphorylation is induced by Salmonella during invasion, and the results found with the phospho-mimic S267A and S276E mutants are not strikingly different. The Myo6+ assay is also a ‘artificial’ as they do not probe a role of Myo6 performed in a regular cell, although they demonstrate an ability of the mutation to change the way Myo6+ can promote filopodia initiation. Thus there is less convincing data in this manuscript regarding the role of this phosphorylation for a particular role of Myo6 in cells.”

In response to reviewer 2's criticism of the Salmonella invasion experiments, this data now has been removed from the manuscript and we state that the experiments using MYO6+ are a cell-based assay and any conclusions on the physiological relevance have been toned down or removed from the manuscript.

“• In discussion, instead of summarizing the unresolved questions from the results, the authors describe in general terms possible sequential roles of Myo6 in microvilli but no reference is provided. “

We apologize for the lack of references and now several references are included to document the importance of MYO6 for endocytosis from the apical domain of polarised epithelial cells: (Ameen et al. 2007; Hegan et al. 2012; Swiatecka-Urban et al. 2004). Growing evidence suggests a role for MYO6 in movement of receptors down a microvillus, which is summarised in Crajoinas et al. 2019.

“Is it clear that no endocytosis occurs at the tip of microvilli and that this is only found at the base?”

Yes, this is a well-documented fact. A reference showing by electron microscopy the distribution of clathrin-coated vesicles at the apical domain of polarised epithelial cells is now included: Shurety et al. 1996.

“A statement indicating what type of experiments could be an approach to test the role of phosphorylation in cell processes in future studies would be important. There is currently no clear evidence for a kinase that would be efficient for the phosphorylation of Ser267 or a consequence in the lack of this kinase for different cellular roles of Myo6. It would be important that the authors acknowledge that there is currently no evidence for the need of Myo6 phosphorylation to perform these functions.”

A statement highlighting that further experiments are required to test the role of DYRK2 in for example clathrin-mediated endocytosis is now included in the discussion:

“The capacity of MYO6 to transition between different modes of activity regulated by S267 phosphorylation/dephosphorylation may be important for its function at the apical domain of polarised epithelial cells⁴⁵⁻⁴⁷. In these cells a dimeric complex of MYO6 and Dab2 has been suggested to facilitate the movement of receptors and exchangers to allow clustering at the base of microvilli^{48 49}. This process might require MYO6 phosphorylation to generate a fast transport mechanism. At the base of the microvilli, where clathrin-coated vesicles are clustering⁵⁰, the MYO6/Dab2 complex assumes a role in receptor uptake through clathrin-coated vesicles, which may require

MYO6 to perform tethering or holding functions in its dephosphorylated state. However, at the present time the impact of MYO6 S267 phosphorylation on clathrin-mediated endocytosis is not known and further experiments are required to test the importance of DYRK2 in regulating this process. These could involve using DYRK2 KO cell lines, DYRK2 inhibitors or overexpressing DYRK2 in polarised epithelial cells.”

“• In the view of this reviewer, for consistency, it is also important to clarify the hypotheses regarding the role in cancer (as the manuscript describes that the kinase involved in Ser267 is overexpressed in cancer). “

Although there is a growing list of studies highlighting not only the role of DYRK2 but also MYO6 in cancer, statements referring to this now have been removed from the manuscript.

“• Currently, the reader may wonder whether the study of S267E describes an ‘interesting’ mutant indicating that Myo6 properties could differ by a mutation in this place, rather than a study of the role of phosphorylation of myo6 required for distinct specific functions / cellular roles of myo6 in distinct places/life time of the cell. The authors should be more clear and modest in results/discussion for what they can conclude from their cellular assays. There is a tendency to ‘sell’ the potential role of an interesting potential modulation without much evidence for this to be really occurring in cells. “

All statements regarding a physiological impact of S267 phosphorylation have been removed or toned down.

“Major detailed comments regarding this point :

2) The assays at a cellular level are not convincingly indicating that Myo6 uses phosphorylation to perform a particular cellular function. The fact that only 1-2 % of Myo6 is phosphorylated, whatever the stimulation with EGF or PMA makes it unclear whether the phosphorylation is really important for cancer cells, unlike what is proposed/indicated in the manuscript when the link of DYRK2 with cancer is mentioned.”

Please also see earlier statement regarding level of phosphorylation in this rebuttal letter. The time MYO6 S267 phosphorylation is required during a specific step in a cellular pathway may be very short lived. However, MYO6 S267 phosphorylation has not only been identified in our phosphoproteomic experiments, but is also recorded in several publications summarised on PhosphoSitePlus.

As we are not testing a synergistic role of MYO6 and DYRK2 in cancer in this manuscript, all statements regarding cancer have been removed from the abstract, introduction, results and discussion.

“The current manuscript is in fact having only limited data for indicating that the motor gets phosphorylated in cells, and thus needs to study a phospho mutant, which is not the best way to indicate why a cellular function would require a distinct cellular function controlled by phosphorylation. Data from a mutant indicating that the lack of phosphorylation leads to the apparition of a phenotype in cell would be more convincing.”

Please see comments in previous sections of the rebuttal letter.

“Additional evidence for the role of the phosphorylation in cancer cell survival, proliferation or migration would be required to keep the text that mentions cancer

roles of DYRK2.

In sum, there is not a clear demonstration of how Myo6 is regulated by phosphorylation and which function would be affected if the residue would be unable to be phosphorylated (S267A mutant)."

All statements regarding cancer have been removed from the abstract, introduction, results and discussion. In addition, the experiments on Salmonella invasions have been removed from the manuscript and any conclusions on the cellular consequences of MYO6 S267 phosphorylation been deleted.

"3) In addition, the phosphorylation performed with DYRK2 in vitro from FL Myo6 or fragment of Myo6 would be expected to be efficient if Myo6 is a substrate of this kinase, which is not the case. This is also demonstrating the poor efficiency of the phosphorylation event.

Overall, there is a doubt that this phosphorylation plays a role in cells and the authors mainly focus on studying phospho-mimic mutants, which limits the study."

We are surprised that the reviewer states that DYRK2 does not phosphorylate the MYO6 peptide or the full-length protein efficiently. In figure 2 d the MYO6 peptide is phosphorylated with same efficiency by recombinant DYRK2 as the Dyrktide, which is the artificial "perfect" substrate for this kinase. Overall, the readout for the in vitro phosphorylation of the full length MYO6 is lower than the peptide, but still significantly higher than the controls. This assays only allows to assess level of phosphorylation relative to a positive control, which is available for the peptide, but not for the full-length MYO6.

"4) Line 292-294: it is not clear whether the chimera MYO6+ works under no load in these assays and whether this conclusion can be suggested.

"Taken together, these results suggest that the increased speed and/or decreased time this mutant spends attached to actin filaments allows MYO6+^{S267E} to outcompete MYO6+^{WT} at growing filopodia tips." Indeed, we don't know whether the MYO6+^{WT} or the MYO6+^{S267E} are attached to a cargo and therefore work under load, however, this would apply to both the wildtype and phosphomimetic MYO6, as the only difference between the two constructs is a single point mutation in the motor domain and the tail domain, important for cargo binding, is exactly the same.

The filopodia assays indicate that the phosphorylation-mimic mutant increases the number of filopodia but this is difficult to reconcile with the role Myo6+ has since the chimera is not described and it is thus difficult to understand how this artificial motor promotes filopodia and why phosphorylation might impact its localisation and function."

Please see below our response on the description of the MYO6+ chimera, which can be found in detail in the reference: Masters and Buss, PNAS 2017.

"5) Salmonella entry assay and Figure 5i. The assay does not seem to be robust as the data points show large discrepancy, among different assays performed with the mutant S267A.

Moreover, the role of Myo6 in Salmonella entry should be better summarized based on previous studies/hypothesis. Is it possible to propose why the phosphorylation of the motor would influence the entry of Salmonella ?"

The Salmonella data have now been removed from the manuscript.

“6) It is important to explain specifically how stiffness is measured and how it compares to other motors as the authors indicate that the phosphorylation event is modulating this parameter. It might be a better choice in discussion to provide a clear statement of the changes in the properties of the motor that would result from phosphorylation and if this would make the motor more performant on all cellular tasks / or more performant only on some of them”.

The design of the stiffness measurements and data analysis are now described both in the main text (line 301-341) and in the methods section (line 795-800).

We have included a comparison of the stiffness measurements in this manuscript with previous studies on MYO5 and MYO2 (skeletal and smooth muscle), taking the different modes of motor attachment to the experimental chamber into account (line 530-550).

“7) Discussion: it is important to summarize the kinetics and the model for describing Myo6 processivity and duty ratio that were previously proposed by other groups. In particular, how the assays performed here compared to WT Myo6 studies in the Sweeney lab would be informative. It seems that ATP binding was also previously described to be the rate limiting step and the model allowing control of gating between heads for Myo6 was proposed to differ from that of Myo5. This should be discussed here. Also, what was measured as stall force for Myo6 or Myo5 should be added line 433-434. See Sweeney et al Embo J 2007. Taking in consideration papers of Myo6 and Myo5, lines 446-448 should be revised.”

We have expanded the discussion of the chemo-mechanical properties of MYO6 and MYO5 accordingly (see above, line 420-427).

“8) Discussion of different hypotheses on the role phosphorylation might have in endocytosis seems to be pure speculation. It would be important to be more precise on what is known about endocytosis in microvilli and to provide references. There is no discussion on how phosphorylation could be triggered, and by which kinase. Instead of hypothetical roles of myo6 in cells for which no kinase is important for regulation, it would be of interest to propose follow up experiments that could describe what are the cellular processes that depend on kinases and Myo6 that could be”

Please see answer above.

“9) be more precise on the mechanism on different cellular roles myo6 may have in cells that do not have any evidence about the importance of a phosphorylation event of Myo6 or a mutation on Ser267 as having a phenotype.”

We don't understand this statement.

“a. Lines 455-458 and the rest of the discussion: there is no strong ground in linking the different roles of Myo6 with the diverse cellular processes “

We don't understand this statement.

“Additional articles that previously indicated the role of phosphorylation for regulation of speed and processivity (Comment 1)

• *The authors do not state correctly the findings of the Robert-Paganin et al. 2019 article. They only mention the methods.*

We added a more detailed description of the experiments and results described in Robert-Paganin et al (line 427-434; line 443-445).

o Lines 316-317: the sentence used to indicate the results of Robert-Paganin et al. 2019 is not appropriate. Only the methods are described and there is no acknowledgement of the results that this study and already described the possibility to tune a motor for different roles (fast motility / load resistance) depending on phosphorylation.

See above; we have now described and discussed this paper on Malaria myosin in more detail in the context of our work (lines 427-434).

o Lines 493-496 : Previous studies on PfMyoA have indicated that a single phosphorylation site regulates the duty ratio of the motor and that this phosphorylation for the invasion of the parasite. This study (Robert Paganin et al) should be cited as a publication establishing the importance of phosphorylation in controlling motor function. And 'for the first time' should be omitted as this PfMyoA study as well as those previously mentioned for phosphorylation of the HCM loop in the control of other myosins should be mentioned in discussion.

Additional phosphorylation sites (comments 2,3, 4)"

see above; 'for the first time' in our manuscript refers to the fact that we have measured the change in stiffness of a single acto-myosin crossbridge as the motor proceeds through the actin-attached states of the crossbridge cycle. From these single molecule experiments we can deduce the force in piconewton a single motor can generate as it proceeds through the actin-attached states and relate the kinetics in our measurements for ADP-release (linked to the second step of the working stroke) and ATP-binding (following the second step of the working stroke) to the different states of stiffness revealed using a forcing function.

"• The authors have corrected errors and improved the manuscript and the changes are satisfactory."

Thank you.

• The authors do not reveal clearly if their analysis of the proteome lead to the identification of other potential Myo6 phosphorylation sites. The authors could cite their mass spec results, without having to identify the consequences of each of these mutations. However, it is of interest to depict whether other phosphorylation sites have been previously described elsewhere in the Myo6 motor and if these sites were also phosphorylated in the experiment depending on external cellular signals, in particular since little amount of phosphorylation is found on S267.

As requested by this reviewer in the first revision of our manuscript we now have included T405 as further phosphorylation site identified in our phosphoproteome data set. We also determined the level of T405 phosphorylation in our mass spec data set and showed that T405 is not phosphorylated by DYRK2. Finally, we generated, expressed and purified the MYO6 mutants T405A or E as well as the T405/S267 double mutants, which were used in actin gliding assays to determine the impact on MYO6 velocity. In contrast to S267, mutating T405 to either A or S has no impact on motor velocity.

"Constructs used in these studies (comment 5)

Unlike stated in the rebuttal, the authors do not describe clearly which Myo6 constructs they are purifying and using. They mention the isoform but not the exact length of the construct used. In addition a clear description of the Myo6+ construct should be provided.”

We clearly included in the first revision of our manuscript not only the isoform but also the length (1253 aa) and accession number of the full-length construct used in all experiments. The only exception is the inverted motility assay where a shorter (991 aa) construct with a C-terminal zipper was used. The paragraph in our Material and Methods reads as follows “MYO6 constructs used in this study are the full-length no-insert isoform (1253 aa, UniProtKB/Swiss-Prot: Q9UM54-5, NCBI NP_001355065.1). For the inverted motility assay a zippered dimer MYO6 construct was created by adding a C-terminal GCN4 leucine zipper sequence (MKQLEDKVEELLSKKNYHLEN EVARLKLVGER) after amino acid 991.”

“For example, the constructs used for the transient kinetic assays should be precised in the main text (and in methods). Currently references are mentioned in methods without clearly establishing the constructs used for these assays. In fact, it is important to mention the construct used clearly for all the in vitro / in cell assays.”

We now specify under “Constructs and antibodies” that the same full-length no insert MYO6 expressed in insect cells was used for the following in vitro experiments:

“MYO6 constructs used in this study are the full-length no-insert isoform (1253 aa, UniProtKB/Swiss-Prot: Q9UM54-5, NCBI NP_001355065.1), which was used for the in vitro motility, the solution kinetic assays and the in vitro phosphorylation assays. “

“For example : Line 258-259 : whether the tail is part of the construct is not mentioned – the lever arm extension is not introduced / defined, whether the constructs can dimerize is not clearly mentioned.”

The exact MYO6+ construct is described in detail in Masters and Buss, PNAS 2017, which is cited in the text. However, for clarity we have now included the exact amino acid boundaries in the MYO6+ chimera as follows: “Initially, we utilized the plus end-directed MYO6 mutant, in which the unique reverse gear (insert 2), the IQ motif and the lever arm extension of MYO6 is replaced by the six-IQ-domain lever arm of MYO5 (MYO6(1-770):Myo5(763-909):MYO6(913-end)²⁵ (Fig. 5a).”

“Ca²⁺ and phosphorylation (comment 7)

• Without a clear description of what is known about motility in presence of Ca²⁺ and cargo, it is unclear that what is described in rebuttal is full speculation. The explanations are unclear and speculative. The authors are not linking directly the relation that would occur between amount of Ca²⁺ and phosphorylation and should state what they propose more clearly.”

Please see below.

“Myo6+ and filopodia results (comment 8)

• The authors have taken into account the previous modifications requested. The fact that the S267E mutant and WT would not lead to mix motors is still unclear that the authors did not explain. It would be of interest to know what are the differences between the constructs, besides the mutation.”

Our imaging data are clear evidence that MYO6S267E and MYO6 WT labelled with two different fluorophores do not overlap in localisation and our quantification confirms the lack in colocalization with a low Pearson’s coefficient (see figure 5f). Therefore,

there is no indication that the WT and MYO6 S267E do form dimers, which would lead to colocalization. The WT and the MYO6 S267E construct are exactly the same besides the mutation.

“Invasion assays (comment 9)

- The authors have taken into account the previous modifications requested. The comment about figure 4g was missing not : “The data presented in 4g does not seem to show ... So this comment was confusing as written.*
- However, the fact that the difference is not so strong between the A and E mutant is not in favour of indicating a strong phenotype depending on Myo6 phosphorylation, see comment 1, which does not lead to a good assay to support the significance of this potential phosphorylation site.”*

These data now have been removed from the manuscript.

“• The title should not only indicate that phosphorylation regulates nucleotide exchange, but also indicate whether this exchange is increased/decreased “

The title now has been changed accordingly.

“• Please indicate in methods and in the legend of figures which constructs allowed to perform the different assays. This is not clear for example for the kinetic studies. Thus, it is difficult to interpret the data, in particular the role of Ca²⁺ in modulating ADP release.”

As detailed above in all experiments the full-length no insert MYO6 was used.

“• Line 72-73 : one recent study published on Myo6 specifically address the distinct role of cargo-adaptor proteins in motor activation and dimerization and should be added here : Canon et al. 2023.”

This reference has been added.

“• Line 81-82 : similarly, this question was addressed by Canon et al and should be mentioned in introduction or discussion.”

Canon et al. demonstrated the importance of GIPC for release of autoinhibition and proximal tail dimerization, however, does not address how the diverse functions of MYO6 are coordinated. Therefore, we do not feel it is the right reference for this paragraph.

“• Line 127 : why use an alphafold structure of Myo6 to show where S267 since there is actual data of the motor : high resolution structures of Myo6 ?”

We prefer to show the whole MYO6 and the overall domain organisation, which is available in AlphaFold.

“• Line 132 : mention to a figure is missing to state that the levels of phosphorylation is 1-2%”

This has now been included on page 5.

“• Line 155 : the graph 2f is a bit misleading since it only show the top of the graph. It would be important to represent the phosphorylation also as done in figure 1f.”

In figure 1f we also show the numerical values in 1g, which allows the reader to fully assess the results. Although the overall phosphorylation at S267 is very small, after PMA stimulation phosphorylation more than doubled. To be able to visualise these

small changes in graph form in figure 2f, we focus on the top part of the graph and the y-axis clearly shows the values between 90 and 100 %.

“• *Line 195: mutant is missing after MYO6S267A-mutant*”

“for both the MYO6^{S267A}- (17.4 ± 0.7) and the MYO6^{S267E}-mutant (18.2 ± 0.7)”.

Happy to receive guidance from the editorial team whether it should read: “for both the MYO6^{S267A}-mutant (17.4 ± 0.7) and the MYO6^{S267E}-mutant (18.2 ± 0.7)”

• *Line 195 : add nm for the working stroke*

done

• *Line 209 / 246 / 248 and elsewhere – the rates are mentioned as mM-1 rather than μM-1*

We think that the numbers describing these rates are easier to read when given in mM-1 s-1

• *Line 214 : W/S 2 is not a state but a transition*

Line 205-209 now reads: Ensemble-averaging analysis³⁴ enabled us to link k_1 and k_2 to specific mechanical events during a single motor's crossbridge cycle. In this experiment, k_1 is determined by the time required to complete both substeps of the W/S, while k_2 characterises the time between W/S 2 and detachment (Fig. 4a, c).

• *Line 217: 34 is not the right publication, structure of Myo6*

Line 211: we have now cited Wells et al. 1999

• *Lines 236-240: Please introduce why testing the role of Ca²⁺ in transient kinetics is of interest.*

We now have given an additional rationale on testing the role of Ca²⁺ in the transient kinetics results section as follows:

“MYO6 has calmodulin light chains on the lever arm and so there is a question if calcium binding to the calmodulin plays any regulatory/modulatory role either as part of the on-off switch with cargo or via a change in the mechanical properties of the lever arm. Calcium binding to the first calmodulin of the lever arm of MYO1B and 1C has been reported to alter ADP release in an unloaded motor^{36 37}. Since we have already shown that the phosphomimetic mutation can influence ADP release, it was of interest to examine if there is an interplay between calcium-calmodulin and ADP release. We find that MYO6^{S267E} is inhibited by calcium (3-4 fold) and to a greater extent than MYO6^{WT} or MYO6^{S267A} (less than 50%). How this is linked to the previous reports that calcium plays a role in regulating cargo binding^{18,38,39}, remains to be explored.”

“• *Lines 254-255 or in methods: describe precisely the chimera Myo6+ (which residues of Myo6 and Myo5 are included in the construct). It is not easy to find the information in the references mentioned.*”

We have now included in the manuscript a summary of the amino acids from MYO6 and MYO5 that are present in the MYO6+ construct on page 8. In the paper by Masters and Buss, PNAS 2017, we describe in great detail how the MYO6+ chimera was generated, which is available in the online Material and Methods. Extract from our PNAS paper:

“Cloning of MYO6+. MYO6+ was constructed by replacing the unique insert (reverse gear), IQ domain and lever arm extension of human myosin VI (gene ID 4646, UniProtKB Q9UM54-5, corresponding to isoform 5 containing no alternative splicing inserts) with the six IQ domain lever arm from mouse myosin V (Myo5, gene ID17920). MYO6+V2 was constructed by replacing the unique insert (reverse gear), and IQ domain with the first two IQ domains of the myosin V lever arm. The six IQ domains of Myo5 were cloned into GFP-MYO6-NI (NI indicates the No Insert isoform) (40) by overlap extension PCR. The tail region of MYO6 was amplified from amino acid 913 to the C-terminal end with sense oligo 5'-GCGTGAGCTGA GAAACTCAAAttacagaaaaaaaac agcagg-3' (the uppercase 5' end corresponds to the 3' end of the 6IQ region of Myo5 at amino acid 909) and antisense oligo 5'-TTTgcgggccgcTTATTTCAA- CAGGTTCTGCAGC-3' (containing a NotI site). The Myo5 motor domain and lever arm (amino acids 1–909) was amplified with sense oligo 5'-aaagaattctgatggctgctgagctctacaca-3' and antisense 5'-cctgctgttttttctgtaaTTTGAGTTTCTTCAGCTCACGC-3' (with the uppercase section corresponding to the reverse complement of the 3' Myo5 lever arm sequence). These two products were then combined via an overlap extension reaction to create a Myo5(S1)-Myo6(Tail) recombinant fusion construct. To create MYO6+, the Myo5 motor domain in Myo5(S1)-Myo6(Tail) was replaced by the MYO6 motor domain by a further round of overlap extension PCR. The motor domain of MYO6 (amino acids 1–770) was amplified with sense oligo 5'-caagaattcaaattggaggatggaaagccc-3' and antisense oligo 5'-GGCAGCCCGAAGTTTGTTCAGCcatgatctgataaattctgc-3', where the uppercase region corresponds to the reverse complement of the 5'-end of the Myo5 lever arm. The Myo5 lever arm and MYO6-tail fusion were amplified by sense oligo 5' gcagaatttgatcagatcatgGCTGAC AAACCTTCGGGCTGCC-3' and antisense oligo 5'-TTTgcgggccgcTTATTTCAACAG GTTCTGCAGC-3' (as used in the first round). The two products were then fused by overlap extension to create the final construct: MYO6(1-770):Myo5(763-909):MYO6(913-end), abbreviated as MYO6+. This was further subcloned into the widely used transient expression vector pEGFP-C3 such that a GFP tag was present at the N terminus.”

“• *Line 299: ‘recently’ please provide a reference”*

The Salmonella data have now been removed from the manuscript.

• *Line 316-317: the sentence is awkward. It is unclear why the authors introduce the reference 40 and 41 only to cite the reference as a method, rather than for citing the results describing that PfMyoA motility modulated by phosphorylation. This was asked in the previous review and the rebuttal indicate that it was corrected but this is not the case.*

see above

“• *Line 354 : the experiment performed is a phospho-mimic (mutation) rather than a phosphorylation and it might be safer to use “phosphorylation at a single site may lead to”*

We now have added extra supplementary data showing the molecular dynamic simulation not only for the phosphomimetic but also for the phosphorylated S267. We observe a very similar reduction in stability for insert-1 in the phosphorylated S267 and phosphomimetic mutant. Thus, the title the results section on MDS does not need to be changed.

“Line 365-366 : the PDB chosen is that of a mutant form of Myo6 - Please add reference here and comment in methods where the main chain of the starting model of the simulation after relaxation differs from that of the WT rigor myosin. The use of the mutant PDB 4DBP as a starting model is not justified and it seems odd to use structure factors to restrain the structure. The method used by the authors and the simulation results should be clarified. It would be important to show also that the minimization has reached convergence and describe better what are the consequences seen in the flexibility of the region near the phosphorylation and elsewhere in the motor.”

We thank the reviewer for the comment. We have now added a reference for the mutant MYO6 and updated the methods. We have selected the mutant crystal structure due to its better geometry compared to the WT MYO6 structures (2BKH, 2BKI). Here is the validation output (obtained using the MolProbity server) for 2BKH, 2BKI and 4DBP:

Analysis output: all-atom contacts and geometry for 2bkhH.pdb

Summary statistics

All-Atom Contacts	Clashscore, all atoms:	3.65	99 th percentile* (N=331, 2.40Å ± 0.25Å)
	Clashscore is the number of serious steric overlaps (> 0.4 Å) per 1000 atoms.		
Protein Geometry	Poor rotamers	51	6.28% Goal: <0.3%
	Favored rotamers	695	85.59% Goal: >98%
	Ramachandran outliers	1	0.11% Goal: <0.05%
	Ramachandran favored	883	96.61% Goal: >98%
	Rama distribution Z-score	-2.27 ± 0.23	Goal: abs(Z score) < 2
	MolProbity score	1.98	94 th percentile* (N=8058, 2.40Å ± 0.25Å)
	CB deviations >0.25Å	0	0.00% Goal: 0
	Bad bonds:	3 / 7574	0.04% Goal: 0%
Peptide Omegas	Bad angles:	2 / 10189	0.02% Goal: <0.1%
	Cis Prolines:	0 / 27	0.00% Expected: ≤1 per chain, or ≤5%
Additional validations	Chiral volume outliers	0/1112	
	Waters with clashes	10/285	3.51% See UnDowser table for details

In the two column results, the left column gives the raw count, right column gives the percentage.

* 100th percentile is the best among structures of comparable resolution; 0th percentile is the worst. For clashscore the comparative set of structures was selected in 2004, for MolProbity score in 2006.

^ MolProbity score combines the clashscore, rotamer, and Ramachandran evaluations into a single score, normalized to be on the same scale as X-ray resolution.

Key to table colors and cutoffs here: ?

Analysis output: all-atom contacts and geometry for 2bkiH.pdb

Summary statistics

All-Atom Contacts	Clashscore, all atoms:	2.38	100 th percentile* (N=97, 2.90Å ± 0.25Å)
	Clashscore is the number of serious steric overlaps (> 0.4 Å) per 1000 atoms.		
Protein Geometry	Poor rotamers	80	9.89% Goal: <0.3%
	Favored rotamers	631	78.00% Goal: >98%
	Ramachandran outliers	10	0.96% Goal: <0.05%
	Ramachandran favored	979	94.41% Goal: >98%
	Rama distribution Z-score	-4.47 ± 0.20	Goal: abs(Z score) < 2
	MolProbity score	2.16	98 th percentile* (N=3760, 2.90Å ± 0.25Å)
	CB deviations >0.25Å	0	0.00% Goal: 0
	Bad bonds:	2 / 8136	0.02% Goal: 0%
Peptide Omegas	Bad angles:	2 / 10993	0.02% Goal: <0.1%
	Cis Prolines:	0 / 27	0.00% Expected: ≤1 per chain, or ≤5%
Low-resolution Criteria	CaBLAM outliers	25	2.4% Goal: <1.0%
	CA Geometry outliers	4	0.39% Goal: <0.5%
Additional validations	Chiral volume outliers	0/1227	
	Waters with clashes	2/49	4.08% See UnDowser table for details

In the two column results, the left column gives the raw count, right column gives the percentage.

* 100th percentile is the best among structures of comparable resolution; 0th percentile is the worst. For clashscore the comparative set of structures was selected in 2004, for MolProbity score in 2006.

^ MolProbity score combines the clashscore, rotamer, and Ramachandran evaluations into a single score, normalized to be on the same scale as X-ray resolution.

Key to table colors and cutoffs here: ?

Analysis output: all-atom contacts and geometry for 4dbpH.pdb

Duke Biochemistry
Duke University School of Medicine

Summary statistics

All-Atom Contacts	Clashscore, all atoms:	2.93	100 th percentile* (N=456, 2.20Å ± 0.25Å)
	Clashscore is the number of serious steric overlaps (> 0.4 Å) per 1000 atoms.		
Protein Geometry	Poor rotamers	9	1.10% Goal: <0.3%
	Favored rotamers	781	95.71% Goal: >98%
	Ramachandran outliers	1	0.11% Goal: <0.05%
	Ramachandran favored	903	98.26% Goal: >98%
	Rama distribution Z-score	-0.73 ± 0.24	Goal: abs(Z score) < 2
	MolProbity score [^]	1.12	100 th percentile* (N=10167, 2.20Å ± 0.25Å)
	Cβ deviations >0.25Å	0	0.00% Goal: 0
	Bad bonds:	0 / 7684	0.00% Goal: 0%
Bad angles:	0 / 10341	0.00% Goal: <0.1%	
Peptide Omegas	Cis Prolines:	0 / 27	0.00% Expected: ≤1 per chain, or ≤5%
Additional validations	Chiral volume outliers	0/1130	
	Waters with clashes	20/781	2.56% See UnDowser table for details

In the two column results, the left column gives the raw count, right column gives the percentage.

* 100th percentile is the best among structures of comparable resolution; 0th percentile is the worst. For clashscore the comparative set of structures was selected in 2004, for MolProbity score in 2006.

[^] MolProbity score combines the clashscore, rotamer, and Ramachandran evaluations into a single score, normalized to be on the same scale as X-ray resolution.

Key to table colors and cutoffs here: ?

This is the validation report of our model before energy minimization:

Analysis output: all-atom contacts and geometry for 4dbp_refinement_refine_1FH.pdb

Duke Biochemistry
Duke University School of Medicine

Summary statistics

All-Atom Contacts	Clashscore, all atoms:	5.98	98 th percentile* (N=456, 2.20Å ± 0.25Å)
	Clashscore is the number of serious steric overlaps (> 0.4 Å) per 1000 atoms.		
Protein Geometry	Poor rotamers	5	0.60% Goal: <0.3%
	Favored rotamers	744	89.96% Goal: >98%
	Ramachandran outliers	2	0.21% Goal: <0.05%
	Ramachandran favored	932	97.69% Goal: >98%
	Rama distribution Z-score	-1.27 ± 0.24	Goal: abs(Z score) < 2
	MolProbity score [^]	1.39	99 th percentile* (N=10167, 2.20Å ± 0.25Å)
	Cβ deviations >0.25Å	0	0.00% Goal: 0
	Bad bonds:	0 / 7753	0.00% Goal: 0%
Bad angles:	0 / 10450	0.00% Goal: <0.1%	
Peptide Omegas	Cis Prolines:	0 / 27	0.00% Expected: ≤1 per chain, or ≤5%
Low-resolution Criteria	CaBLAM outliers	7	0.7% Goal: <1.0%
	CA Geometry outliers	3	0.32% Goal: <0.5%
Additional validations	Chiral volume outliers	0/1146	
	Waters with clashes	24/781	3.07% See UnDowser table for details

In the two column results, the left column gives the raw count, right column gives the percentage.

We added a sentence in methods stating the RMSD between the starting model of the simulation (after relaxation) and the WT (PDBID:2BKH) rigor myosin.

The steepest descent algorithm has converged after 2834, 3131 and 2561 steps in the WT, S267E and PhosphoS267 systems, respectively. We have attached all energy minimisation log files (EM_WT.log, EM_S267E.log and EM_S267Phospho) for the reviewer.

“• Line 381 : 1 nm apart : please indicate the atoms from which this measure is taken and how it differs in WT and S267E”

We have removed this text. Fig. 7b, d now clearly show the conformational change occurred in the insert-1 domain.

“• Lines 411-414 : sentences are unclear.what are the additional phosphorylation events referred to here ? Only these two are mentioned – are there any other found in PhosphoSitePlus ?”

This reviewer requested in the first round of revisions to highlight the limitations of our study. Therefore, we have included in the discussion potential additional phosphorylation sites, which may be identified in the future and may also play a role in regulating MYO6 motor activity. The screenshot at the beginning of this rebuttal letter shows curated data for MYO6 on PhosphoSitePlus. While S267 and T405 are listed with the highest number of publications in the motor domain, other phosphorylated sites have been identified with lower frequency.

“• Line 460 Note that a recent publication indicated that GIPC does not induce multimerization but dimerization of Myo6 (Canon et al Nat Comm 2023). “

This part of the manuscript has been removed.

Reviewer #3 (Remarks to the Author):

As discussed with the editor I am only able to make limited comments on the experimental methods that use optical tweezers. The three-bead assay as used here was pioneered by some of the authors. As in the previous submission I find these methods to be well and clearly described, with good quality figures to illustrate both methods and results.

The authors have corrected the repeated citations of the previous versions.

REVIEWERS' COMMENTS

Reviewer #2 (Remarks to the Author):

The authors have removed weak parts of the paper to focus on very strong results which are now well presented.

The characterization of the role of the phosphorylation and its impact on motor function is now convincing. I fully support publication of these results.

Just a comment regarding the review sent last time : I am sorry to see that the review was long and repetitive due to the fact that the rebuttal had not been provided, when i first was given this new version of the article. I had to review the paper twice, the second time in order to take into account the rebuttal. Both reviews were then provided to the authors, while they were redundant.

I congratulate the authors for their nice findings.